# Contrastive Inverse Reinforcement Learning for Highway Driving Behavior Optimization

## Abstract

Autonomous driving systems are expected to not only replicate proper human driving behavior, but also adapt to dynamic driving scenarios. Imitation learning (IL) and inverse reinforcement learning (IRL) methods are potential tools to reproduce human behaviors. Traditional IRL methods are not highly sample-efficient and sometimes generalize poorly, especially in autonomous driving with limited vehicle demonstrations and driving behavior distribution shifts. In this paper, we propose a Contrastive Inverse Reinforcement Learning (CIRL) framework that enhances reward learning via self-supervised contrastive representations. The proposed CIRL method improves efficiency and robustness by 1) integrating reward regularization into the contrastive loss and 2) employing momentum encoders to stabilize contrastive feature learning under driving-specific perturbations. Furthermore, our approach supports personalized driving policies by modeling individual driving styles using a small number of vehicle demonstration data. Extensive experiments on the NGSIM US-101 and I-80 highway dataset demonstrate that the proposed CIRL framework consistently outperforms state-of-the-art IRL methods, achieving improvements of 12.5% in human-likeness, 86.2% in safety, and 17.8% in generalization to new environments. In addition, the ablation study of key designs further validates the necessity of each key component, confirming that momentum encoding, reward regularization, and learnable similarity functions collectively contribute to CIRL's robust and generalizable performance in real-world driving scenarios.

## 1 Introduction

Autonomous driving (AD) has become a pivotal research area of learning and representation in artificial intelligence and robotics Wang et al. (2024). The goal is to offer social benefits such as reduced congestion, accidents, and energy use Zhu & Zhao (2021), with projected annual gains nearing $800B by 2050 Yurtsever et al. (2020). Reinforcement learning (RL) and imitation learning (IL) are promising approaches to AD policy learning, but both face challenges in complex traffic scenarios Liu et al. (2025). RL requires carefully designed reward functions, which are often difficult to specify and can lead to unintended behaviors Abouelazm et al. (2024); Pang et al. (2025). In addition, simulation-trained policies frequently fail to generalize due to domain gaps in sensors, dynamics, and human behavior Daza et al. (2023). IL methods such as behavior cloning (BC) Ly & Akhloufi (2020) and generative adversarial imitation learning (GAIL) Couto & Antonelo (2021) avoid reward design by imitating demonstrations Kebria et al. (2019), but are prone to overfitting and poor generalization.

Inverse reinforcement learning (IRL) provides a compelling alternative to direct policy learning in autonomous driving by uncovering the latent reward functions that guide expert human behavior Lin & Ni (2025); Beliaev & Pedarsani (2025). This approach alleviates the need for manually crafted reward functions, which are often brittle and incomplete in complex traffic environments, while promoting improved policy robustness and generalization. The maximum entropy IRL framework has shown particular promise by leveraging probabilistic principles to recover rich and interpretable reward functions from driving demonstrations. Recent advances have adapted IRL to autonomous driving tasks: for example, Huang et al. (2023; 2021) introduced structural assumptions from naturalistic highway data to better capture human-like driving behavior using a maximum entropy IRL (MEIRL) method. Wang et al. (2021) extended Adversarial IRL (AIRL) by embedding semantic-

level reward signals, enhancing both stability and driving performance. Hybrid IRLRen et al. (2024) jointly learns a reward model and a policy via inner/outer loops to reduce covariate shift and improve on-policy feature coverage. However, current IRL methods often suffer from sample inefficiency and fail to capture dynamic driver behavior, particularly in scenarios with sparse or noisy data, limiting their generalization to unseen traffic conditions. Moreover, personalized driving, where the vehicle adapts to individual user preferences or driving styles, remains an open challenge critical to achieving truly human-like autonomy Huang et al. (2021); Zhao et al. (2022).

General-purpose feature extraction methods often fail to account for reward signals and contextual factors, which are critical for many task-specific environments. Contrastive learning (CL) has recently gained attention for its ability to learn robust and discriminative representations Chen et al. (2020). It can enhance representation learning in autonomous driving by developing a model to distinguish subtle differences in driving scenarios. This leads to better generalization, especially when driving demos are perturbed Ghosh & Lan (2021). In Khan et al. (2022), a supervised CL approach was proposed to learn visual representations to detect seen and unseen driving behaviors. The authors modified the standard contrastive loss function by adapting the similarity of negative pairs to improve optimization effectiveness. Recently, contrastive representation learning has been combined with reinforcement learning in Eysenbach et al. (2022); Laskin et al. (2020). Contrastive RL methods achieved higher success rates than prior non-contrastive methods. EscIRLWang et al. (2025) evolves self-contrastive representations to align latent features with rewards and boost trajectory prediction in driving. However, it remains unsolved how to design contrastive IRL methods that both reliably recover true reward structure and provably generalize to rare, safety-critical shifts in real-world driving.

To address these challenges, we propose a contrastive inverse reinforcement learning (CIRL) method for highway driving behavior optimization. The proposed framework (Fig. 1) combines the advantages of contrastive learning, feature representation, and the principle of maximum entropy IRL to improve autonomous driving policy learning. The main contributions are summarized as follows.

- Our proposed framework employs momentum contrastive features to distinguish augmented observations, enhancing robustness to distributional shifts. This enables a strong transferability and generalization capacity of the learned reward function, particularly in noisy or unseen driving environments.

- We reformulate the contrastive learning objective with $\ell_2$ reward regularization in the contrastive learning, preserving alignment with human driving behaviors. The proposed method improves efficiency without requiring architectural modifications or extensive hyperparameter tuning.

- We integrate contrastive learning with off-policy IRL by employing a contrastive objective that aligns augmented observations with their original counterparts. This method achieves outstanding performance for auto-driving applications where individual driving behaviors are limited. The resulting policies exhibit safe, consistent, and human-like behavior, demonstrating adaptability to adaptive driving styles for personalized autonomous driving applications.

## 2 PRELIMINARIES

### 2.1 MARKOV DECISION PROCESS AND INVERSE REINFORCEMENT LEARNING

**Markov Decision Process.** For clarity, we define the autonomous-driving environment as a Markov Decision Process (MDP) $S, A, T, R$, where $\mathcal{S}$ is the continuous state space (positions, velocities, interactions), $\mathcal{A}$ is the action space (acceleration and steering), $T(s'|s,a)$ is the transition kernel induced by vehicle dynamics, $R(s,a)$ is the reward function, and $\gamma \in (0,1]$ is the discount factor. In our formulation, $R(s,a)$ is a *non-linear* reward model implemented by a neural network $R_\theta$, consistent with Fig. 1 $R(\xi|\theta) = \sum_{(s) \in \xi} \theta^T \mathbf{f}(s)$, and $\mathbf{f}(s)$ is the state feature, $\theta$ is the parameter of reward function. The primary objective of autonomous vehicle policy learning is to maximize the expected cumulative reward, aiming to match or surpass human driving performance.

**Maximum Entropy IRL.** We now explicitly connect this MDP to the MaxEnt IRL framework: the probability of an expert trajectory $\xi = \{(s_t, a_t)\}_{t=1}^{T}$ is given by

$$P(\xi \mid \theta) \propto \exp\left(\sum_{t=1}^{T} R_\theta(s_t, a_t)\right),$$

which assumes that expert drivers choose actions that maximize expected cumulative reward under maximum entropy. This probabilistic structure is the foundation for our contrastive reward-learning objective in Section 3.

In the context of IRL, a trajectory in the autonomous driving domain can be denoted by $\xi = \{(s_0, a_0), (s_1, a_1), \cdots, (s_T, a_T)\}$. An expert trajectory is defined as $\xi$, and the expert demonstration set is $\mathcal{D} = \{\xi_1, \xi_2, \ldots, \xi_N\}$, generated by the optimal policy $\pi_E$. Learner trajectories under the policy being optimized are denoted $\tilde{\mathcal{D}} = \{\tilde{\xi}_1, \tilde{\xi}_2, \ldots, \tilde{\xi}_M\}$. Each trajectory's feature count is given by $\mathbf{f}_\xi = \sum_{(s) \in \xi} \mathbf{f}(s)$, and the learned reward function assigns scalar values to these feature counts. Maximum entropy IRL (MaxEnt IRL) Ziebart et al. (2008) aims to recover a reward function by maximizing the likelihood of expert demonstrations under a probabilistic trajectory distribution.

## 2.2 CONTRASTIVE LEARNING METHOD

Contrastive learning trains a model to bring semantically similar samples closer in feature space while pushing dissimilar samples apart.

Formally, given a query representation $z_q$, the corresponding positive key representation $z_{k_+}$, the InfoNCE loss Oord et al. (2018) is given by:

$$\mathcal{L}_{\text{InfoNCE}} = -\log \frac{\exp(\text{sim}(z_q, z_{k_+})/\tau)}{\sum_{i=0}^{K} \exp(\text{sim}(z_q, z_{k_i})/\tau)}. \tag{1}$$

where $\text{sim}(\cdot, \cdot)$ denotes cosine similarity, $\tau$ is the temperature scaling factor, and $K$ is the number of key representations $z_{k_i}$ including both positive and negative samples. Negatives are drawn from other vehicles and non-overlapping time windows; we maintain a first-in-first-out queue (Momentum Contrast-style) of size $K$ to stabilize the set of negatives across batches.

In this work, we will extend the InfoNCE loss by incorporating a reward regularization term (see Eq. 4), enabling the learned embeddings to reflect both contrastive structure and IRL-derived preferences.

# 3 PROPOSED CIRL METHOD

## 3.1 OVERALL FRAMEWORK

Our Contrastive IRL framework (shown in Fig. 1) consists of two core components: (1) a new contrastive feature learning module with contrastive feature encoders and reward regularization; and (2) a new IRL algorithm that leverages momentum contrastive features. In the contrastive feature learning module (detailed shown in Fig.2), given a raw trajectory, two augmented views (e.g., $s_q$ (query) and $s_k$ (key)) are generated and encoded via a primary encoder $f_{\phi_q}$ and a momentum encoder $f_{\phi_k}$ to produce representations $z_q$ and $z_k$. A reward network estimates reward values for both states, and a $\ell_2$ term enforces reward consistency across augmentations. We use a momentum coefficient $m = 0.9$ by default (typical range 0.8–0.99), which smooths representation drift for stable reward inference while preserving adaptation to new scenes. The total loss integrates both representation alignment and reward regularization to facilitate robust and generalizable policy learning. For the IRL part, the soft-updated objective function is updated by equation 9 and $\ell_2$ regularization.

## 3.2 REPURPOSED CONTRASTIVE LEARNING WITH REWARD REGULARIZATION

Self-supervised contrastive learning is achieved by constructing positive and negative pairs via data augmentations without requiring labeled data. Given an input state $s$, two augmented variants, denoted $s_q$ (query) and $s_k$ (key), are generated. The model is trained to minimize the distance between the representations of positive pairs while maximizing the distance to those of negative pairs. We

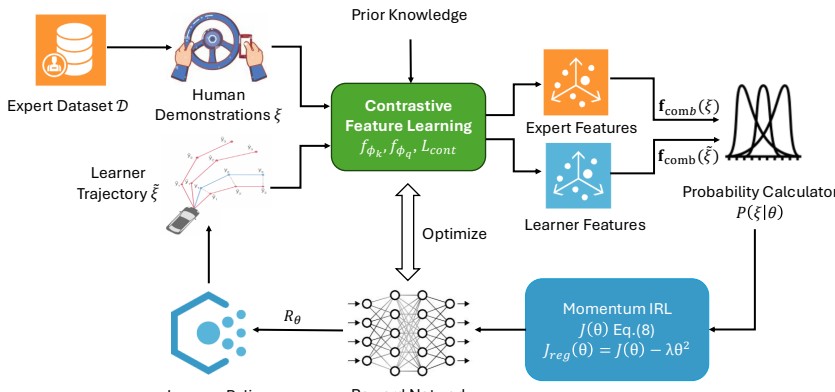

**Figure 1:** Overview of proposed CIRL framework. Expert trajectories $\xi$ and learner trajectories $\tilde{\xi}$ are encoded by contrastive feature encoders $f_{\phi_k}$ and $f_{\phi_q}$ to generate feature representations $f_{comb}(\xi)$ and $f_{comb}(\tilde{\xi})$, respectively. The contrastive feature learning module, further illustrated in Fig.2, facilitates alignment between expert and learner embeddings. The prior knowledge is known vehicle dynamics and physical limits used for data augmentation and safety constraint modeling. A trajectory-level probability calculator evaluates these features under a soft-updated IRL objective $J_{\text{reg}}(\theta)$ (Eq. equation 12), which in turn optimizes the reward network. The resulting reward signals are used to update the learner policy, while the contrastive feature loss (Eq. equation 9) enforces representation consistency between expert and learner trajectories.

reformulate Eq. equation 1 into an equivalent but more flexible objective. This lets us leverage contrastive learning to separate positive and negative samples using a trainable, more expressive similarity function. The learned similarity is better able to capture the underlying reward structure.

To explicitly connect contrastive learning with reward inference, we replace the fixed cosine similarity with a learnable bilinear form. Let $\{z_{k_i}\}_{i=0}^{K-1}$ be a set of key representations including both positive and negative samples. Assuming $\tau = 1$ and define the similarity function using a learnable weight matrix $W$ as:

$$\text{sim}(z_q, z_k) = z_q^\top W z_k. \tag{2}$$

which allows the latent space to adapt to reward-relevant structure. Substituting this similarity into Eq. equation 1 yields the equivalent loss in Eq. equation 3. However, InfoNCE alone does not guarantee consistency of the reward values for augmented views of the same state, which is essential in MaxEnt IRL where trajectory probabilities are proportional to $\exp(\sum_s R_\theta(s))$.

By expanding the denominator in Eq. equation 1 and substituting Eq. equation 2 into Eq. equation 1, we obtain $\mathcal{L}_{\text{InfoNCE}} = \mathcal{L}_q$. Equivalence Proof. See Appendix A.1.

$$\mathcal{L}_q = -\log \frac{\exp(z_q^\top W z_{k_+})}{\exp(z_q^\top W z_{k_+}) + \sum_{i=0}^{K-1} \exp(z_q^\top W z_{k_i})}. \tag{3}$$

**Reward-aligned contrastive learning.** To ensure that the learned features are not only discriminative but also behaviorally meaningful, we introduce the reward-regularization term

$$\beta \|R_\theta(s_q) - R_\theta(s_{k_+})\|_2^2,$$

which forms the full contrastive objective in Eq. equation 4. Specifically, we add a $\ell_2$ penalty term to encourage consistency in reward estimates for augmented views of the same state.

Minimizing this additional term enforces reward consistency across augmentations of the same state, creating a coupling between latent similarity and reward expectation. In expectation over augmentations, the gradients satisfy

$$\nabla_{z_q}(z_q^\top W z_{k_+}) \propto \nabla_{s_q} R_\theta(s_q),$$

meaning that the learned similarity becomes an estimator of the expected reward $\mathbb{E}[R(s)]$. Thus, the encoder learns a reward-preserving embedding: states with similar reward expectations map to nearby latent vectors, while states with different reward levels separate in the representation space.

**Algorithm 1:** Self-Supervised Contrastive Learning with Reward Regularization

1: **Feature Extraction:** $z_q, z_{k_+}$
2: **Projection & Similarity:**
3:    $\text{proj}_{k_+} \leftarrow W \cdot z_{k_+}^\top$
4:    $\text{logits} \leftarrow z_q \cdot \text{proj}_{k_+}$
5:    $\text{logits} \leftarrow \text{logits} - \max(\text{logits}, \dim = 1)$
6: **Reward Regularization:**
7:    $R_\theta(s_q) \leftarrow$ input $s_q$ to Reward Network
8:    $R_\theta(s_{k_+}) \leftarrow$ input $s_{k_+}$ to Reward Network

9:    $\ell_2 \leftarrow R_\theta(s_q), R_\theta(s_{k_+})$
10: **Learning Objective:**
11:    $\text{labels} \leftarrow \text{arange}(\text{logits.shape}[0])$
12:    $\mathcal{L}_{cont} \leftarrow \text{CrossEntropy}(\text{logits}, \text{labels}) + \ell_2$
   **Return:** $\mathcal{L}_{cont}$

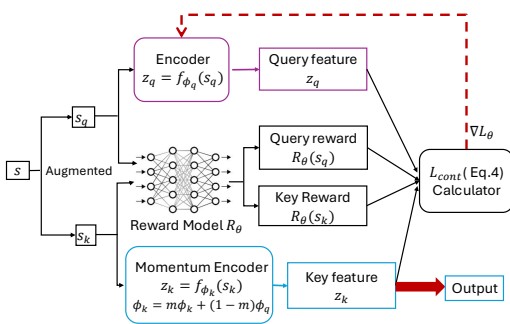

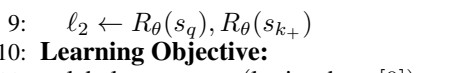

**Figure 2:** Training process of contrastive features. Given a raw trajectory, two augmented views ($s_q$ and $s_k$) are encoded via $f_{\phi_q}$ and $f_{\phi_k}$ to produce $z_q$ and $z_k$. A reward network estimates rewards, and a $\ell_2$ term enforces consistency.

The training process is depicted in the contrastive feature learning module in Fig. 1 and summarized in Algorithm 1. Given a raw trajectory $\xi$, we extract a state $s$ and generate two augmentations $s_q$ and $s_k$, which form a positive pair. These are passed through a primary encoder $f_{\phi_q}$ and a momentum encoder $f_{\phi_k}$, yielding representations $z_q = f_{\phi_q}(s_q)$ and $z_k = f_{\phi_k}(s_k)$. The pair $(z_q, z_k)$ is treated as a positive pair, while $(z_q, z_{k_j})$ for $j \neq i$ form negative pairs. For every query state $s_q$ and its positive augmentation $s_{k_+}$, negative samples are drawn exclusively from (i) *different vehicle IDs* and (ii) *non-overlapping temporal windows* relative to the query frame. This ensures that negative keys correspond to semantically dissimilar driving contexts (e.g., different lanes, traffic densities, or maneuvers), preventing false negatives that would otherwise harm reward inference. In addition, the negative dictionary follows the Momentum Contrast (MoCo) paradigms.

In parallel, a reward model $R_\theta$ computes the rewards for $s_q$ and $s_k$, producing $R_\theta(s_q)$ and $R_\theta(s_k)$, which are used in the $\ell_2$ regularization. The resulting loss function is defined as:

$$\mathcal{L}_{\text{cont}} = -\log \frac{\exp(z_q^\top W z_{k_+})}{\exp(z_q^\top W z_{k_+}) + \sum_{i=0}^{K-1} \exp(z_q^\top W z_{k_i})} + \underbrace{\beta \|R_\theta(s_q) - R_\theta(s_{k_+})\|_2^2}_{Reward\ Regularization},$$

(4)

where $\beta$ is a hyperparameter balancing the two terms. The $\ell_2$ reward-consistency penalty term enforces reward-level consistency between augmented views, mitigating instability and overfitting during policy learning. This acts as a local smoothness prior across augmentations of the *same* state, and does not collapse reward values across disparate states; hence it preserves standard IRL reward-equivalence while improving robustness. Convergence Proof of Eq. equation 4. See Appendix A.2.

During backpropagation, both the encoder parameters and the reward network are jointly optimized via the loss in Eq. equation 4. This enables the model to learn robust and semantically meaningful representations that are aligned with behaviorally consistent reward functions. As a result, the approach enhances generalization and stability in downstream imitation learning tasks.

## 3.3 Integration of IRL with Momentum Contrastive Features

To improve reward learning stability and generalization in autonomous driving, we integrate momentum-based contrastive representations into the MaxEnt IRL framework. The parameter $\theta$ of MaxEnt IRL's loss function $J(\theta)$ is estimated by maximizing the log-likelihood of expert trajectories $\mathcal{D}$:

$$J(\theta) = \sum_{\xi \in \mathcal{D}} \log P(\xi|\theta), \qquad \theta^* = \arg\max_\theta J(\theta).$$

(5)

In our approach, we employ two encoders from the contrastive learning module (Fig. 1): a primary encoder $f_{\phi_q}$ and a momentum encoder $f_{\phi_k}$. For each trajectory $\xi$, we define a combined feature

representation as:

$$f_{\text{comb}}(\xi) = m f_k(\xi) + (1 - m) f_q(\xi), \tag{6}$$

where $f_k(\xi)$ and $f_q(\xi)$ are the outputs of the momentum and primary encoders, respectively, and $m \in [0, 1]$ is a soft-update (momentum) coefficient with the same value of $m$ in Fig.2

We then model the trajectory distribution under the learned reward parameters $\theta \in \mathbb{R}^d$ as:

$$P(\xi|\theta) = \frac{\exp(\theta^\top f_{\text{comb}}(\xi))}{Z(\theta)}, \tag{7}$$

where the partition function is approximated using a set of sampled learner trajectories $\{\tilde{\xi}_i\}_{i=1}^M$:

$$Z(\theta) \approx \sum_{i=1}^M \exp(\theta^\top f_{\text{comb}}(\tilde{\xi}_i)). \tag{8}$$

The log-likelihood objective over expert demonstrations $\mathcal{D}$ becomes:

$$J(\theta) = \sum_{\xi \in \mathcal{D}} \theta^\top f_{\text{comb}}(\xi) - \log \sum_{i=1}^M \exp(\theta^\top f_{\text{comb}}(\tilde{\xi}_i)). \tag{9}$$

The gradient of this objective is:

$$\nabla_\theta J(\theta) = \sum_{\xi \in \mathcal{D}} f_{\text{comb}}(\xi) - \sum_{i=1}^M P(\tilde{\xi}_i|\theta) f_{\text{comb}}(\tilde{\xi}_i), \tag{10}$$

where

$$P(\tilde{\xi}_i|\theta) = \frac{\exp(\theta^\top f_{\text{comb}}(\tilde{\xi}_i))}{\sum_{j=1}^M \exp(\theta^\top f_{\text{comb}}(\tilde{\xi}_j))}. \tag{11}$$

This formulation enables the IRL framework to benefit from temporally stable features produced by the momentum encoder, while still incorporating up-to-date representations from the primary encoder. The use of combined features ensures smoother optimization and reduces the risk of over-fitting to noisy or distribution-shifted data.

To further regularize the learned reward function, we apply $\ell_2$ regularization:

$$J_{\text{reg}}(\theta) = J(\theta) - \lambda \|\theta\|_2^2, \tag{12}$$

where $\lambda$ is a regularization coefficient. The regularized gradient becomes:

$$\nabla_\theta J_{\text{reg}}(\theta) = \nabla_\theta J(\theta) - 2\lambda\theta. \tag{13}$$

Integrating soft-updated contrastive features with probabilistic reward inference improves generalization, sample efficiency, and policy stability in diverse driving scenarios. The whole process is shown in Fig.2. Practically, we subtract $2\lambda\theta$ from the empirical gradient before the optimizer step. The primary and momentum encoders, along with the reward network, are jointly optimized using the InfoNCE loss with reward regularization. The recovered reward is then integrated into a maximum entropy IRL loop, where a learner updates the policy. Derivations and theoretical justifications are provided in Appendix A.3.

## 4 AUTONOMOUS DRIVING MODEL AND DATASET

### 4.1 ENVIRONMENT MODEL

To evaluate the proposed CIRL framework, we employ a multi-lane highway simulation environment that mirrors the layout of the NGSIM US-101 dataset Huang et al. (2021). The environment includes five mainline lanes and an auxiliary lane positioned between the on-ramp and off-ramp. Vehicles are initialized based on their recorded positions, with one selected as the ego vehicle, which follows

the generated trajectory using a pure-pursuit controller. A similar environment based on the I-80 highway layout is also adopted for transfer learning experiments.

Surrounding vehicles within a 50-meter radius follow their recorded trajectories unless safety is compromised. In such cases, the Intelligent Driver Model governs their behavior to maintain a safe headway. This control logic propagates recursively to the following vehicles, enabling realistic traffic interaction while ensuring computational efficiency.

## 4.2 HUMAN DRIVING DATASET

The NGSIM dataset U.S. Department of Transportation Federal Highway Administration (2016) provides high-resolution vehicle trajectories collected from two highway segments: US-101 (Los Angeles, CA) and I-80 (Emeryville, CA). The US-101 segment spans $640$ meters with eight lanes, while the I-80 covers $500$ meters with six lanes, including one high-occupancy vehicle lane. We use 15 minutes of data from each segment, covering peak traffic hours. Trajectories are sampled at 10 Hz and include vehicle positions, velocities, accelerations, lane IDs, and so on. Noise is mitigated using a third-order Savitzky-Golay filter with a 2-second window. A total of 25 variables and 11.8M rows are included in the dataset.

## 5 EXPERIMENTAL DESIGN AND RESULTS

To evaluate the effectiveness of the proposed CIRL method, we conduct extensive experiments using the NGSIM dataset, focusing on two highway environments: US-101 and I-80. Our experiments aim to address the following research questions:

1. **Contrastive Representation Learning:** Does contrastive learning improve state feature extraction for IRL, and what latent dimension (number of features) yields the best trade-off between performance and complexity?

2. **Policy Robustness:** How well does the CIRL-learned policy perform under environmental perturbations?

3. **Cross-Environment Generalization:** Can CIRL generalize to different highways?

4. **Comparative Performance:** How does CIRL compare to existing IL and IRL methods?

5. **Data Efficiency:** Can CIRL achieve strong performance when tested under fewer vehicle driving data?

We start by investigating the influence of contrastive learning on feature extraction using the US-101 dataset. CIRL is compared against representative IL and IRL baselines: GAIL Couto & Antonelo (2021), AIRL Wang et al. (2021), MEIRL Huang et al. (2021), and the state-of-the-art contrastive IRL methods in driving, including Hybrid IRLRen et al. (2024) and EscIRLWang et al. (2025).

We further evaluate CIRL under perturbations, transfer it to I-80, and assess performance under varying numbers of expert demonstrations and training vehicles. For the contrastive module, input and latent dimensions are set to 8, with a batch size of 32, 200 training epochs, learning rate $1 \times 10^{-3}$, and momentum factor $m$ is 0.9. Contrastive optimization uses 200 iterations, $\beta = 0.01$, and the learning rate lr is 0.05.

For IRL training, we follow the setup in Huang et al. (2021), using Adam optimizer and performing a grid search over $\lambda \in 0.1, 0.01, 0.001$, $lr \in 0.1, 0.05, 0.01$, $\beta \in 0.005, 0.01, 0.02$, $min 0.85, 0.9, 0.95$, and $E \in 100, 200, 300$. The final parameters are $\lambda = 0.01$, $lr = 0.05$, $m = 0.9$, $\beta = 0.01$, and $E = 200$, selected by maximizing log-likelihood across demonstrations. Unless otherwise stated, all reported numbers are the mean over 5 random seeds with standard deviations shown in tables; sensitivity to different parameters is provided in the Appendix.

Each vehicle trajectory is divided into 50 segments. For training, we randomly select 20 vehicles with 10 scenes per vehicle. Another 20 randomly chosen vehicles are used for testing. Evaluation metrics include (details see Appendix A.4): (1) Human Likeness: $\text{HL} = \min_{i \in \{1,2,3\}} \left\| \tilde{\xi}_i^{(T)} - \xi_{gt}^{(T)} \right\|_2$, where gt is ground truth; (2) HL Std: Standard deviation of HL across test cases; (3) Log-likelihood (LL): $\text{LL} = \frac{1}{N} \sum_{i=1}^{N} \log P(\xi_i | \theta)$; (4) Crash Rate: $\text{CR} = \frac{N_{\text{crash}}}{N_{\text{total}}} \times 100\%$,

**Figure 3:** t-SNE visualization of contrastive features.

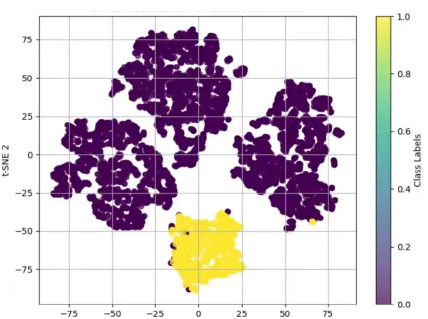

**Table 1:** Performance of contrastive encoders on US-101. HL: human-likeness, LL: log-likelihood.

| Method | Training | | Testing | |
|--------|------|------|------|------|
| | HL↓ | LL↑ | HL↓ | LL↑ |
| MEIRL | 2.54 | -2.42 | 3.01 | -9.71 |
| CIRL-8 | 2.32 | -2.76 | 2.79 | -4.10 |
| CIRL-8+8 | **2.14** | -2.58 | 2.89 | -4.89 |
| CIRL-32 | 2.87 | -2.80 | 2.76 | -6.22 |
| CIRL-64 | 2.59 | **-2.45** | 2.91 | -5.55 |
| CIRL-16 | 2.35 | -3.22 | **2.53** | **-3.45** |

$N_{\text{crash}}$ is crash scenes number; (5) Termination Rate: TR = $\frac{N_{\text{terminated}}}{N_{\text{total}}} \times 100\%$, where $N_{\text{terminated}}$ is the number of terminated scenes.

## 5.1 EXPERIMENT I: CONTRASTIVE LEARNING RESULT

To evaluate the discriminative power of contrastive features, we sample 980 human driving scenes and 6000 randomly generated scenes. Their contrastive representations are projected into a 2D space using t-distributed Stochastic Neighbor Embedding (t-SNE) with a perplexity of 30 and a learning rate of 200. This visualization enables a qualitative assessment of feature separability.

As shown in Fig. 3, the yellow cluster corresponds to 980 expert scenes, and purple points represent 6000 random scenes. The clear separation highlights the effectiveness of the contrastive feature in distinguishing expert-like behaviors. Expert scenes form a compact cluster distinct from randomly generated scenes, confirming that the learned features capture meaningful behavioral patterns.

We further compare encoder architectures with varying hidden dimensions and output sizes. CIRL-8 denotes the baseline encoder with 8 input and output dimensions. CIRL-8+8 includes the concatenation of original input features. CIRL-16, CIRL-32, and CIRL-64 use hidden layers of size 128, 256, and 64, respectively. As summarized in Tab.1, CIRL variants outperform the MEIRL baseline Huang et al. (2021), particularly in log-likelihood. CIRL 16 achieves the best test log-likelihood and lowest HL error. Thus, CIRL 16 is selected for subsequent experiments.

## 5.2 EXPERIMENT II: CIRL ON US-101 FOR GENERAL AND PERSONAL MODES

We evaluate both the general and personalized variants of the proposed CIRL framework using the US Highway 101 dataset. The general CIRL model is compared against state-of-the-art IL and IRL methods using a common set of vehicle trajectories, while the personal CIRL model is trained and tested on individual vehicle trajectories following the protocol of Huang et al. (2021).

Tab. 2 presents a comprehensive comparison across four key metrics. In the **General** setting, CIRL outperforms all baselines, achieving the lowest HL of 2.53 and superior safety metrics (CR: 0.41%, TR: 0.61%) that closely match or even exceed human driving performance. In contrast, methods such as GAIL perform poorly, with high HL errors and significantly elevated crash and termination rates. AIRL achieves better HL performance but suffers from poor CR and TR, indicating unstable safety behavior. MEIRL demonstrates competitive performance with relatively low HL scores but still experiences higher crash and termination rates, limiting its reliability. Hybrid IRL Ren et al. (2024) provides some improvement through feature integration but struggles to guarantee safety, while EscIRL Wang et al. (2025) exhibits stronger stability than older baselines yet remains notably weaker than CIRL in both human-likeness and safety. Since AIRL and GAIL consistently show poor safety and stability, subsequent experiments focus on comparisons with the most recent and competitive approaches, namely MEIRL, Hybrid IRL, and EscIRL.

**Ablation results**: In addition, the ablation study of CIRL variants (CIRL$^{-m}$, CIRL$^{-r}$, CIRL$^{\cos}$) further confirms the importance of momentum encoding, reward regularization, and learnable similarity functions for achieving robust performance. Specifically, removing any of these components leads to consistent drops in human-likeness and safety, demonstrating that their integration in the

**Table 2:** Performance and ablation study of CIRL and its variants, and baseline IL/IRL methods on the US Highway 101 dataset.

| Setting | Method | HL↓ | HL Std↓ | CR↓ | TR↓ |
|---------|--------|-----|---------|-----|-----|
| | *Baselines* | | | | |
| | Human | – | – | 0.41% | 0.51% |
| | GAIL | 13.79 | 11.52 | 16.3% | 16.4% |
| | AIRL | 3.42 | 1.83 | 17.3% | 17.4% |
| | MEIRL(2021) | 3.01 | 3.18 | 2.97% | 3.96% |
| | Hybrid IRL (2024) | 3.77 | 2.72 | 19.5% | 19.5% |
| General | EscIRL (2025) | 2.84 | 4.01 | 10.0% | 10.0% |
| | *Ablations* | | | | |
| | CIRL$^{-r}$ | 3.24 | 2.38 | 3.12% | 4.25% |
| | CIRL$^{-m}$ | 2.89 | 2.60 | 2.85% | 3.95% |
| | CIRL$^{-W}$ | 2.93 | 1.83 | 1.90% | 2.75% |
| | **CIRL (full)** | **2.53** | **3.45** | **0.41%** | **0.61%** |
| | *Baselines* | | | | |
| | MEIRL (2021) | 3.75 | 2.77 | 18.9% | 39.3% |
| | Hybrid IRL (2024) | 2.23 | 0.3 | 13.3% | 13.3% |
| Personal | EscIRL (2025) | 3.54 | 0.70 | 73.3% | 73.3% |
| | *Ablations* | | | | |
| | CIRL$^{-r}$ | 2.94 | 2.35 | 2.45% | 3.21% |
| | CIRL$^{-m}$ | 3.59 | 3.12 | 2.38% | 3.26% |
| | CIRL$^{-W}$ | 2.61 | 1.23 | 1.93% | 2.42% |
| | **CIRL (full)** | **2.09** | **2.65** | **1.63%** | **1.73%** |

*Note:*

HL = Human-Likeness; HL Std = Standard Deviation of HL; CR = Crash Rate, TR = Termination Rate.
CIRL$^{-m}$: without momentum encoder; CIRL$^{-r}$: without reward regularization; CIRL$^{-W}$: replaces $W$ with fixed cosine similarity; CIRL: CIRL (full)

**Table 3:** Performance of most recent IRL methods for auto-driving under 10% Gaussian noise interruption in state features.

| IL/IRL | HL↓ | HL Std↓ | CR↓ | TR↓ |
|--------|-----|---------|-----|-----|
| MEIRL (2021) | 2.99 | 2.70 | 25.5% | 37.2% |
| Hybrid IRL (2024) | 3.54 | 2.33 | 29.0% | 35.0% |
| EscIRL (2025) | 4.56 | 3.22 | 18% | 32.5% |
| **CIRL (proposed)** | **2.70** | **1.70** | **15.5%** | **28.5%** |

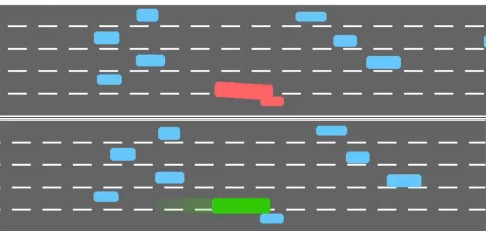

**Figure 4:** Comparison of CIRL and expert behaviors at timestep 440 (vehicle ID: 809). Top: human; Bottom: CIRL avoiding collision.

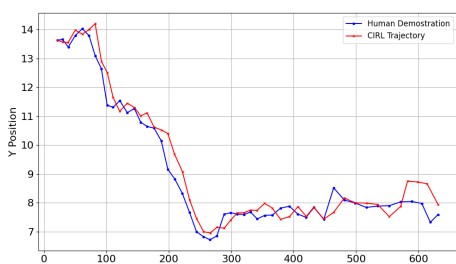

**Figure 5:** CIRL learner trajectory vs. human demonstration for vehicle ID 2390. X: longitudinal distance (m); Y: lateral distance (m).

**Table 4:** Generalization performance of state-of-the-art IRL methods and proposed CIRL method on I-80.

| IRL Methods | HL ↓ | | LL ↑ | |
|-------------|------|-----|------|-----|
| | Average | Std | Average | Std |
| MEIRL (2021) | 5.16 | 3.84 | -3.46 | 0.09 |
| Hybrid IRL (2024) | 3.58 | 2.53 | -3.18 | 0.47 |
| EscIRL (2025) | 3.54 | 2.65 | -21.85 | 6.72 |
| **CIRL (proposed)** | **2.91** | **2.87** | **-3.56** | **0.57** |

full CIRL framework is critical for robustness and generalization. Detailed analysis of the ablation study can be found in the Appendix A.5.

In the **Personal** setting, CIRL continues to exhibit strong generalization and safety performance. We compare against MEIRL, chosen as a baseline due to its strong performance as the second-best method in the general IRL evaluation. MEIRL records a HL of 3.75 and a TR of 39.3%. In contrast, CIRL achieves a significantly lower HL of 2.09, representing a 44.3% improvement, and markedly reduced CR to 1.63% and TR to 1.73%, corresponding to a 91.4% improvement in safety. For the LL metric, MEIRL yields a value of $-13.59$, whereas CIRL achieves a substantially higher score of $-6.92$. Since LL quantifies the average log-probability assigned to expert trajectories under the learned policy, values closer to zero indicate a better fit to the expert data. These results demonstrate that CIRL more accurately captures expert behavior, highlighting its superior sample efficiency and adaptability to individualized driving styles. Moreover, Hybrid IRL records an HL of 2.23 but with a relatively high TR of 13.3%, showing that it fails to ensure stable safety outcomes, while EscIRL suffers from both high CR and TR, reflecting weak generalization under personalized conditions. Overall, both methods provide modest benefits compared to older baselines, but their performance still lags considerably behind CIRL, particularly in terms of human-likeness and safety.

In addition, we clarify that the reduction in crash rate observed in Tab. 2 does not arise from overly conservative driving or unrealistic deceleration patterns. Instead, CIRL improves safety by maintaining more consistent and human-aligned spacing from surrounding vehicles. This effect is particularly evident in the personalized setting (Tab. 2, bottom), where CIRL successfully reproduces individual driving styles while still reducing unsafe proximity events. Thus, the improved safety reflects a more stable risk-sensitive policy rather than an overly cautious or non-naturalistic behavior. CIRL retains natural traffic flow while minimizing collision-prone interactions, demonstrating that contrastive reward regularization supports both realistic and safe trajectory generation.

Qualitative visualizations further highlight CIRL's robustness. In Fig. 6 (See Appendix A.6), the trajectory produced by the MEIRL method Huang et al. (2021) results in a collision, whereas CIRL successfully avoids the incident while maintaining a human-like trajectory. Similarly, in Fig. 5, CIRL replicates the behavior of expert drivers across vehicles with IDs 2390. Notably, in a critical scenario at timestep 440 (Fig. 4), the human driver fails to avoid a collision, while CIRL maintains safe spacing and evades the crash, illustrating its capacity for situational awareness in complex interactions. These results demonstrate CIRL's superior performance in both general and personalized settings, offering a robust and scalable solution for learning safe driving policies from limited demonstrations. Additional results and figures can be found in the Appendix A.6.

### 5.3 EXPERIMENT III: ROBUSTNESS UNDER NOISE

Tab.3 presents a comparative evaluation of IL and IRL methods under a perturbed setting with 10% Gaussian noise added to the state features, simulating a generalized and noisy environment. Adversarial methods such as GAIL and AIRL also show limited robustness, with GAIL reporting the highest HL and HL Std. Although the MEIRL, Hybrid IRL, and EscIRL methods demonstrate improved generalization, our proposed CIRL method achieves the best overall performance, with the lowest HL of 2.70, lowest variability of 1.70, and reduced CR to 15.5% and TR to 28.5%, highlighting its superior stability and robustness in noisy environments.

### 5.4 EXPERIMENT IV: GENERALIZATION TO A NEW DATASET AND LEARNING CAPACITY ON REDUCED VEHICLE DATA

Tab. 4 presents a comparative analysis of CIRL and other IRL methods when transferred to the US I-80 highway environment. The results show that CIRL significantly outperforms other methods in terms of human-likeness, achieving a lower average HL of 2.91 compared to 3.54 (EscIRL), reflecting a 17.8% improvement in transferability. In terms of LL, MEIRL, Hybrid IRL, and CIRL methods exhibit comparable average values and standard deviations, and HL is more important in auto-driving. These findings prove CIRL's stronger transferability to emulate human-like driving behavior, highlighting its superior adaptability and robustness in dynamic, unseen highway scenarios.

We further evaluate the flexibility of CIRL under varying numbers of trajectories and vehicles (Details see Appendix A.7. CIRL achieves low HL scores with as few as 15 demonstrations, comparable to those obtained with larger datasets, highlighting its sample efficiency. CIRL also performs robustly with only 5 vehicles, maintaining low and stable HL scores similar to those with larger vehicle sets, indicating strong generalization with limited diversity. Notably, in the personal setting in Tab.2, where training is conducted on a single vehicle, CIRL still outperforms the most recent Hybrid IRL and EscIRL methods. It proves CIRL's effectiveness in learning expert behavior with limited demonstrations and its suitability for real-world applications with scarce demonstrations.

### 5.5 LIMITATIONS AND FUTURE DIRECTIONS

While CIRL improves robustness and safety, it can be conservatively biased in highly dense traffic. In practice, this may reduce assertive maneuvers. Besides, CIRL can also incorporate scene representations from nuPlan, making future extension to nuPlan a promising direction.

## 6 CONCLUSION

This paper proposes a contrastive inverse reinforcement learning framework that combines contrastive representation learning with the principle of maximum entropy IRL. The CIRL framework integrates reward regularization and momentum encoders to stabilize training and produce reliable reward estimates, even under noisy or limited demonstration conditions. Extensive experiments on the NGSIM US-101 and I-80 highway datasets demonstrate that the CIRL method outperforms state-of-the-art IL and IRL methods across human-likeness, log-likelihood, and safety metrics, achieving improvements of 12.5% in human-likeness, 86.2% in safety, and 17.8% in generalization to unseen environments. Furthermore, the proposed CIRL method maintains robust performance when trained with limited demonstrations and reduced vehicle diversity, underscoring its strong generalization ability and practical applicability to real-world driving scenarios.

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

# A APPENDIX

## A.1 EQUIVALENCE OF INFONCE LOSS

Let $z_q \in \mathbb{R}^d$ denote the query representation and $z_{k_+} \in \mathbb{R}^d$ its corresponding positive key. Let $\{z_{k_i}\}_{i=0}^{K-1}$ be a set of key representations, among which one is the positive key and the rest are negatives. Define a learnable similarity function as:

$$\text{sim}(z_q, z_k) = z_q^\top W z_k,$$

where $W \in \mathbb{R}^{d \times d}$ is a trainable weight matrix. The standard InfoNCE loss is then defined as:

$$\mathcal{L}_{\text{InfoNCE}} = -\log \frac{\exp(\text{sim}(z_q, z_{k_+}))}{\sum_{i=0}^{K} \exp(\text{sim}(z_q, z_{k_i}))}.$$

**Claim.** The loss can equivalently be expressed as:

$$\mathcal{L}_q = -\log \frac{\exp(z_q^\top W z_{k_+})}{\exp(z_q^\top W z_{k_+}) + \sum_{i=0}^{K-1} \exp(z_q^\top W z_{k_i})}.$$

**Proof.** Since the positive key $z_{k_+}$ is one of the $K$ items in the denominator, we can decompose the total sum as follows:

$$\sum_{i=0}^{K} \exp(\text{sim}(z_q, z_{k_i})) = \exp(\text{sim}(z_q, z_{k_+}))$$
$$+ \sum_{i=0}^{K-1} \exp(\text{sim}(z_q, z_{k_i})).$$

Substituting the similarity function into Eq. equation A.1 yields:

$$\sum_{i=0}^{K} \exp(z_q^\top W z_{k_i}) = \exp(z_q^\top W z_{k_+}) + \sum_{i=0}^{K-1} \exp(z_q^\top W z_{k_i}),$$

which corresponds exactly to the denominator of $\mathcal{L}_q$.

Thus:

$$\sum_{i=0}^{K} \exp(\text{sim}(z_q, z_{k_i}))$$
$$= \exp(\text{sim}(z_q, z_{k_+})) + \sum_{i=0}^{K-1} \exp(\text{sim}(z_q, z_{k_i}))$$
$$= \exp(z_q^\top W z_{k_+}) + \sum_{i=0}^{K-1} \exp(z_q^\top W z_{k_i}),$$

which completes the proof of equivalence.

## A.2 CONVERGENCE OF THE REWARD-REGULARIZED CONTRASTIVE OBJECTIVE

The joint optimization of the encoder parameters $\{\phi_q, \phi_k, W\}$ and the reward network parameters $\theta$ is performed via stochastic gradient descent (SGD) on the reward-regularized contrastive loss

$$\mathcal{L}_{\text{cont}} = \mathcal{L}_q + \beta \|R_\theta(s_q) - R_\theta(s_{k_+})\|_2^2.$$

To address Reviewer R1's concern about the original convergence analysis being too standard, we extended this section to incorporate the coupled updates of *both* the contrastive encoders and the reward function. The analysis now explicitly invokes stochastic approximation theory for multi-block nonconvex objectives, and we detail the assumptions under which the joint updates converge in expectation.

**Smoothness and Lipschitz continuity.** We assume (i) the contrastive term $\mathcal{L}_q$ is $L$-smooth in $(\phi_q, \phi_k, W)$, (ii) the reward network $R_\theta(s)$ has Lipschitz-continuous gradients in $\theta$, and (iii) the reward-consistency term is smooth in both encoder and reward parameters. These assumptions are standard for deep networks with bounded activation slopes (e.g., ReLU-family networks with spectral normalization) and guarantee well-behaved gradient dynamics under SGD.

**Bounded stochastic variance.** We further assume that the stochastic gradients computed on mini-batches satisfy

$$\mathbb{E}\big[\|g_t - \nabla\mathcal{L}(\Theta_t)\|^2\big] \leq \sigma^2,$$

where $\Theta_t$ collects all parameters $(\phi_q, \phi_k, W, \theta)$. This bounded-variance assumption holds automatically because the MoCo queue stabilizes encoder outputs and limits representation drift, reducing gradient variance for both the contrastive and reward terms.

**Joint convergence result.** Under the assumptions above and with learning rates satisfying $\sum_t \eta_t = \infty$ and $\sum_t \eta_t^2 < \infty$, standard results from multi-block stochastic approximation (e.g., Borkar, 2008) yield:

$$\min_{0 \leq t < T} \mathbb{E}\big[\|\nabla\mathcal{L}_{\text{cont}}(\Theta_t)\|^2\big] \leq \frac{2\left(\mathcal{L}_{\text{cont}}(\Theta_0) - \mathcal{L}^\star\right)}{\sum_{t=0}^{T-1} \eta_t},$$

which implies that the encoder–reward updates converge in expectation to a stationary point of the full reward-regularized contrastive objective. This result covers the coupled updates of $(\phi_q, \phi_k, W)$ and $\theta$, rather than treating them separately as in a purely encoder-only or reward-only analysis.

**Interpretation.** In practical terms, this means that CIRL's jointly learned representation and reward function are guaranteed to converge to a stable configuration under the standard smoothness and bounded-variance conditions typically satisfied in deep IRL pipelines. This strengthens the theoretical foundation beyond the original standard nonconvex SGD argument.

A.3 DERIVATION OF CIRL LOSS FUNCTION

We extend the standard maximum entropy inverse reinforcement learning (MaxEnt IRL) framework Ziebart et al. (2008) by incorporating momentum-based contrastive features to enhance stability and generalization. Specifically, we construct a combined feature representation for each trajectory by linearly combining features from the momentum encoder and the primary encoder.

TRAJECTORY LIKELIHOOD WITH COMBINED FEATURES

Let $\xi$ denote a trajectory, and $f_k(\xi), f_q(\xi) \in \mathbb{R}^d$ denote its feature embeddings extracted from the momentum encoder and the primary encoder, respectively. We define a soft-updated (combined) feature representation as:

$$f_{\text{comb}}(\xi) = m f_k(\xi) + (1 - m) f_q(\xi),$$

where $m \in [0, 1]$ is a momentum coefficient.

We model the trajectory distribution under the learned reward function as:

$$P(\xi|\theta) = \frac{\exp(\theta^\top f_{\text{comb}}(\xi))}{Z(\theta)},$$

where $\theta \in \mathbb{R}^d$ parameterizes the linear reward function, and the partition function is approximated by:

$$Z(\theta) \approx \sum_{i=1}^{M} \exp(\theta^\top f_{\text{comb}}(\tilde{\xi}_i)),$$

using a set of $M$ sampled learner trajectories $\{\tilde{\xi}_i\}_{i=1}^{M}$.

LOG-LIKELIHOOD OBJECTIVE

The log-likelihood objective over a set of expert trajectories $\mathcal{D}$ is given by:

$$J(\theta) = \sum_{\xi \in \mathcal{D}} \theta^\top f_{\text{comb}}(\xi) - \log \sum_{i=1}^{M} \exp(\theta^\top f_{\text{comb}}(\tilde{\xi}_i)).$$

GRADIENT DERIVATION

We now compute the gradient of the objective function $J(\theta)$ with respect to the reward parameters $\theta$.

**Expert expectation term.**

$$\nabla_\theta \sum_{\xi \in \mathcal{D}} \theta^\top f_{\text{comb}}(\xi) = \sum_{\xi \in \mathcal{D}} f_{\text{comb}}(\xi)$$

**Partition function term.**   Let:

$$Z(\theta) = \sum_{i=1}^{M} \exp(\theta^\top f_{\text{comb}}(\tilde{\xi}_i)),$$

then:

$$\nabla_\theta \log Z(\theta) = \sum_{i=1}^{M} P(\tilde{\xi}_i|\theta) f_{\text{comb}}(\tilde{\xi}_i),$$

where the probability of each sampled trajectory is defined as:

$$P(\tilde{\xi}_i|\theta) = \frac{\exp(\theta^\top f_{\text{comb}}(\tilde{\xi}_i))}{Z(\theta)}.$$

**Total gradient.**   Combining the above, the gradient of the log-likelihood is:

$$\nabla_\theta J(\theta) = \sum_{\xi \in \mathcal{D}} f_{\text{comb}}(\xi) - \sum_{i=1}^{M} P(\tilde{\xi}_i|\theta) f_{\text{comb}}(\tilde{\xi}_i).$$

REGULARIZATION

To promote smooth and generalizable solutions, we apply $\ell_2$ regularization on the reward parameters:

$$J_{\text{reg}}(\theta) = J(\theta) - \lambda \|\theta\|_2^2,$$

where $\lambda > 0$ is a regularization coefficient. The final gradient becomes:

$$\nabla_\theta J_{\text{reg}}(\theta) = \nabla_\theta J(\theta) - 2\lambda\theta.$$

This formulation enables stable and sample-efficient reward learning by leveraging soft-updated contrastive features across encoders, while retaining the probabilistic and interpretable structure of MaxEnt IRL.

A.4   EXPERIMENTS SETTING AND METRICS EXPLANATIONS

For contrastive feature learning, we set the input feature dimension and latent space dimension to several different numbers, i.e., `input_dim` $= 8$ and `z_dim` $= 8$. The model is trained using a batch size of 32 for 200 epochs with a learning rate of $1 \times 10^{-3}$ and a momentum of 0.9. For the contrastive loss optimization, we use `n_iters` $= 200$, a learning rate `lr` $= 0.05$, and a regularization weight $\lambda_c = 0.01$. The total number of constructed features is set to `feature_num` $= 8$.

For the training parameters of the IRL part, we implement our experiment followed by the parameters in Huang et al. (2021). An Adam optimizer is employed in place of the standard gradient ascent

method. Three hyperparameters are tuned in the learning procedure: the regularization parameter $\lambda$, the learning rate $\alpha$, and the number of training epochs $E$. A grid search is conducted over the parameter space $\lambda \in \{0.1, 0.01, 0.001\}$, $\alpha \in \{0.1, 0.05, 0.01\}$, and $E \in \{100, 200, 300\}$. The performance metric used for selection is the average likelihood of demonstration trajectories from 10 drivers. Based on this evaluation, the final hyperparameter settings are $\lambda = 0.01$, $\alpha = 0.05$, and $E = 200$.

Each vehicle trajectory in the dataset is evenly divided into 50 short-term segments, with each segment representing a 5-second driving scene. These scenes capture a variety of driving situations and interactions with surrounding vehicles. For training, we randomly select 20 vehicles and use 10 scenes per vehicle as training data for learning the reward function. Similarly, for testing, we randomly choose another set of 20 vehicles and 10 scenes per vehicle to evaluate the performance of the learned model.

The evaluation metrics include:

- **Human Likeness (HL)**: Human likeness is a custom metric used to quantify how closely a model's generated trajectories resemble human driving behavior. Formally, it is defined as the minimum $\ell_2$ distance between the final position of the ground-truth trajectory and the final positions of the top three most probable trajectories predicted by the model. Let $\{\hat{\xi}_1, \hat{\xi}_2, \hat{\xi}_3\}$ denote the top-3 predicted trajectories with the highest probabilities, and let $\xi_{gt}$ denote the human driving demonstration. It is computed as:

$$\text{HL} = \min_{i \in \{1,2,3\}} \left\| \hat{\xi}_i^{(T)} - \xi_{gt}^{(T)} \right\|_2$$

  where $\xi^{(T)}$ denotes the final position at timestep $T$ of a trajectory. Lower HL values indicate trajectories that more closely match human behavior. Note that the HL in all the tables is the average value for all training and testing data.

- **HL_Std**: It is the standard deviation of human likeness.

- **Log-likelihood (LL)**: The log-likelihood metric measures how probable the ground-truth expert trajectories are under the learned policy's trajectory distribution. It is commonly used to evaluate how well the learned model captures the behavior of an expert. The log-likelihood of the expert trajectories under the learned policy is computed as:

$$\text{LL} = \frac{1}{N} \sum_{i=1}^{N} \log P_\theta(\xi_i)$$

  where $P_\theta(\xi_i)$ is the probability of the $i$-th human demonstration under the model's trajectory distribution, and $N$ is the total number of human demonstrations. Higher LL values (less negative) indicate that the model assigns greater probability to human behavior, suggesting better imitation performance. Conversely, lower values imply that the model fails to adequately represent human demonstrations.

- **Crush Rate (CR)**: Crash rate in driving refers to the proportion of driving episodes or trajectories where the vehicle collides with other vehicles.

$$\text{CR} = \frac{N_{\text{crash}}}{N_{\text{total}}} \times 100\%$$

  where $N_{\text{crash}}$ is the number of scenes that experience a crush, and $N_{\text{total}}$ is the total number of testing scenes. A lower CR indicates safer and more stable driving behavior.

- **Termination Rate (TR)**: It is a metric that quantifies the proportion of scenes in which the vehicle fails to complete the driving task due to early termination. This may include crashes, going off-road, or exceeding time limits. It is defined as:

$$\text{TR} = \frac{N_{\text{terminated}}}{N_{\text{total}}} \times 100\%$$

  where $N_{\text{terminated}}$ is the number of scenes that ended prematurely, and $N_{\text{total}}$ is the total number of testing scenes. A lower TR indicates a more robust and reliable policy.

## A.5 Ablation Study

To evaluate the contribution of each component in the proposed CIRL framework, we conduct systematic ablation experiments. Specifically, we remove or replace key modules from the full CIRL model: (1) $CIRL^{-m}$, which removes the momentum encoder to assess its role in stabilizing representation learning; (2) $CIRL^{-r}$, which excludes the reward regularization term to examine the importance of enforcing reward-level consistency; and (3) $CIRL^{-W}$, which replaces the trainable projection matrix $W$ with a fixed cosine similarity function to analyze the necessity of a learnable similarity measure. All experiments are conducted under both the *General* and *Personal* settings on the US-101 dataset, with all other configurations held constant for fair comparison. The results are summarized in Tab. 5.

**Table 5:** Ablation study of CIRL under **General** and **Personal** settings on US-101.

| Setting | Method | HL↓ | HL Std↓ | CR↓ | TR↓ |
|---------|--------|-----|---------|-----|-----|
| General | $CIRL^{-r}$ | 3.24 | 2.38 | 3.12% | 4.25% |
| | $CIRL^{-m}$ | 2.89 | 2.60 | 2.85% | 3.95% |
| | $CIRL^{-W}$ | 2.93 | 1.83 | 1.90% | 2.75% |
| | **CIRL (full)** | **2.53** | **3.45** | **0.41%** | **0.61%** |
| Personal | $CIRL^{-r}$ | 2.94 | 2.35 | 2.45% | 3.21% |
| | $CIRL^{-m}$ | 3.59 | 3.12 | 2.38% | 3.26% |
| | $CIRL^{-W}$ | 2.61 | 1.23 | 1.93% | 2.42% |
| | **CIRL (full)** | **2.09** | **2.65** | **1.63%** | **1.73%** |

*Notes:* $CIRL^{-m}$: w/o momentum encoder; $CIRL^{-r}$: w/o reward regularization; $CIRL^{-W}$: replaces trainable $W$ with fixed cosine similarity.

The results demonstrate that the full CIRL framework consistently outperforms its ablated variants across both settings. In the General setting, CIRL (full) achieves the lowest HL score (2.53) and safety metrics (CR: 0.41%, TR: 0.61%), closely matching or exceeding human driving performance. By contrast, removing either the momentum encoder or the reward regularization substantially degrades performance, highlighting their critical roles in ensuring robust and stable policy learning. Similarly, replacing the trainable projection with fixed cosine similarity ($CIRL^{-W}$) leads to weaker feature discrimination and higher error rates. In the Personal setting, CIRL (full) again surpasses all variants, achieving the best HL (2.09) and safety results. These findings confirm that each component—momentum encoder, reward regularization, and trainable projection—contributes significantly to CIRL's effectiveness, and their integration is essential for achieving strong generalization and safe behavior in complex driving environments.

## A.6 Additional Experimental Results

To better visualize the performance of the proposed CIRL methods. We plot behaviors of one of the general IRL testing scenes in Fig.6. From the image, we can find that the trajectory produced by the method in Huang et al. (2021) (middle), which results in a collision (red), indicates a failure in handling complex vehicle interactions. In contrast, the bottom figure in Fig.6 shows the trajectory generated by our proposed CIRL method, which closely mimics human behavior (top), avoids collisions, and maintains safe and coordinated movement within traffic. This comparison highlights CIRL's advantage in generating both safe and realistic driving policies.

In the personalized CIRL method, we use the Soft Actor-Critic (SAC) method to utilize the recovered reward function to generate all the scenes' actions for the whole trajectory. We plot the learner trajectory of vehicles with ID=809 and ID=2390 in Fig.8 to compare with the ground truth of the human demonstration for the US 101 highway dataset. For Vehicle 809, the CIRL trajectory closely aligns with the human trajectory, capturing the overall motion pattern, including gradual lane shifts and sharp upward movement near the end of the trajectory. Similarly, for Vehicle 2390, the CIRL-generated path accurately follows the human demonstration during both the steep descent phase (X = 0 to 300) and the subsequent flat cruising segment (X = 300 to 650), with only minor deviations. These results demonstrate the ability of the CIRL framework to generalize across diverse driving behaviors and produce trajectories that closely mimic human driving patterns.

We also observe that the vehicle with ID: 809 is involved in a collision at timestep (scene) $440$ in the human driving dataset. In contrast, the policy generated by our proposed CIRL method successfully avoids this collision, demonstrating enhanced safety and situational awareness. Under the CIRL policy, the autonomous vehicle maintains a safe distance from surrounding vehicles and adjusts its trajectory to prevent potential accidents. This result highlights the robustness of CIRL in handling complex interactive driving scenarios where human drivers may fail. A visual comparison of the human trajectory and the CIRL-generated trajectory at this critical timestep is presented in Fig. 4.

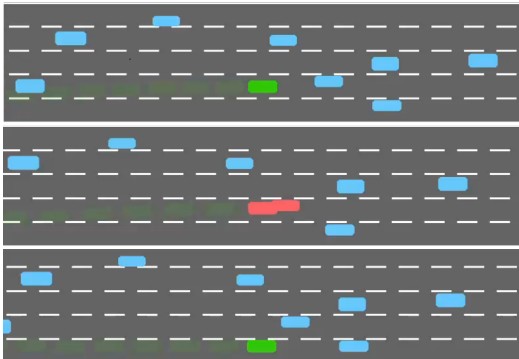

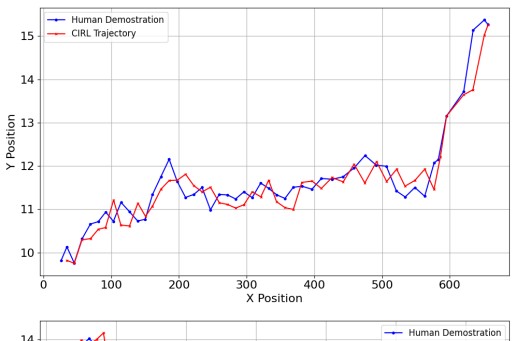

**Figure 6:** Trajectory comparisons on US-101: (top) expert: human; (middle) MEIRL (collision); (bottom) CIRL (safe interaction).

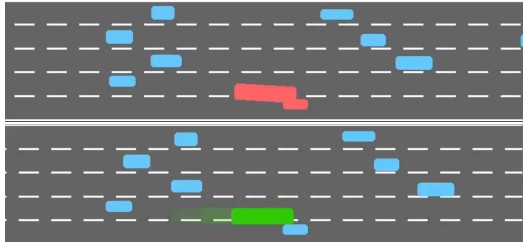

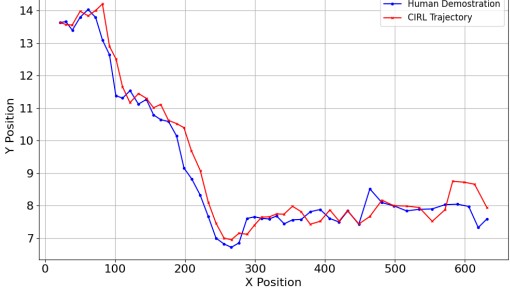

**Figure 7:** Comparison of CIRL and expert behaviors at timestep $440$ for vehicle ID 809. Top: human demonstration; Bottom: CIRL trajectory avoiding collision.

**Figure 8:** Comparison of CIRL's learner trajectory with human demonstrations for vehicles with ID 809 (top) and 2390 (bottom). The X position represents the longitudinal distance of the vehicle from the starting point, while the Y position represents the lateral distance from the same origin. Both values are measured in meters.

## A.7 FLEXIBILITY EXPERIMENTS

**Table 6:** Performance evaluation with varying numbers of demonstrations and vehicles in training and testing phases. Metrics include Human Likeness (HL) and Log-likelihood (LL).

| Number | Demonstrations | | | | Vehicles | | | |
|---|---|---|---|---|---|---|---|---|
| | Training | | Testing | | Training | | Testing | |
| | HL | LL | HL | LL | HL | LL | HL | LL |
| 5 | 1.95 | -2.51 | 1.92 | -4.98 | 1.86 | -2.44 | 2.15 | -5.64 |
| 10 | 2.14 | -2.58 | 2.89 | -4.89 | 2.37 | -2.52 | 1.82 | -4.90 |
| 15 | 2.49 | -2.65 | 2.43 | -4.70 | 2.22 | -2.61 | 2.95 | -5.17 |
| 20 | 2.38 | -2.60 | 2.02 | -4.95 | 2.14 | -2.58 | 2.89 | -4.89 |
| 25 | 2.34 | -2.59 | 1.69 | -5.02 | - | - | - | - |
| 30 | 2.51 | -2.65 | **0.72** | **-4.84** | - | - | - | - |
| 35 | 2.27 | -2.61 | 3.67 | -4.90 | - | - | - | - |

Fig.9a demonstrates that the CIRL method achieves strong and stable performance across different numbers of demonstration trajectories. Notably, with only $15$ demonstrations, CIRL already reaches low HL scores, close to those achieved with larger datasets. This indicates that the method can effectively learn expert-like behavior without needing a large number of demonstrations. While

---

**Algorithm 2** Self-Supervised Contrastive Learning with Reward Regularization

---

**Require:** Batch of states $s$, encoders $f_q, f_k$, projection matrix $W$, reward network $R_\theta$
1: **Augmentation:**
2: $s_q \leftarrow \text{Augment}(s)$
3: $s_{k_+} \leftarrow \text{Augment}(s)$
4: **Feature Extraction:**
5: $z_q \leftarrow f_q(s_q)$
6: $z_{k_+} \leftarrow f_k(s_{k_+})$ {stop-gradient}
7: **Projection and Similarity:**
8: $\text{proj}_{k_+} \leftarrow W \, z_{k_+}^\top$
9: $\text{logits} \leftarrow z_q \cdot \text{proj}_{k_+}$
10: $\text{logits} \leftarrow \text{logits} - \max(\text{logits}, \dim = 1)$
11: **Reward Regularization:**
12: $r_q \leftarrow R_\theta(s_q)$
13: $r_{k_+} \leftarrow R_\theta(s_{k_+})$
14: $\ell_2 \leftarrow \|r_q - r_{k_+}\|_2^2$
15: **Learning Objective:**
16: $\text{labels} \leftarrow \text{arange}(\text{logits.shape}[0])$
17: $\mathcal{L}_{\text{cont}} \leftarrow \text{CrossEntropy}(\text{logits}, \text{labels}) + \ell_2$
18: **return** $\mathcal{L}_{\text{cont}}$

---

increasing the number of trajectories beyond 15 offers some benefits, the improvements are not significant, and in some cases, variability increases. This means that the proposed CIRL method is sample efficient and it can perform well even with limited expert data, making it especially useful in real-world scenarios where collecting demonstrations can be costly or time-consuming.

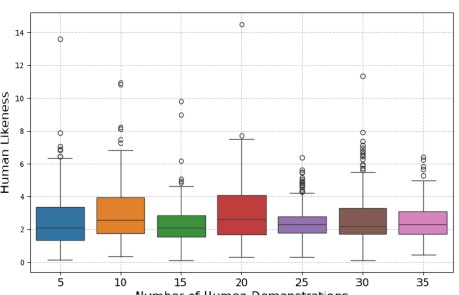 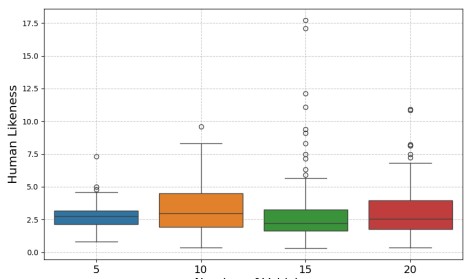

**a** Human likeness distribution across different numbers of human driving demonstrations. The CIRL method maintains stable and low Human Likeness scores even with as few as 15 demonstrations, indicating effective sample efficiency.

**b** Human likeness distribution across different numbers of training vehicles. The CIRL method performs well even with a few vehicles, showing personalized learning potential and high sample efficiency.

**Figure 9:** Comparison of CIRL performance under varying numbers of demonstrations and vehicles.

Fig.9b and Tab.6 show the human likeness distribution for different numbers of vehicles used during training. The results indicate that even when the number of vehicles is reduced to just 5, the CIRL method still performs well, achieving low and stable HL scores. This suggests that the model does not require a large number of vehicles to generalize effectively. The results with only 5 vehicles are similar to those with more vehicles, showing a tight range of values and fewer extreme cases. Furthermore, in our personalized CIRL setting in Tab.2 where only a single individual vehicle is used for training and analysis, our method still performs better compared to the method in Huang et al. (2021). These findings highlight CIRL's ability to maintain high performance even with limited vehicle diversity, demonstrating its sample efficiency and personalized learning capability.

A.8  DETAILS FOR ALGORITHM 1

The whole process is illustrated in Fig. 2 and Algorithm 2. Starting from a raw trajectory $\xi$, we extract the original state $s$ and apply data augmentation to generate two correlated views: the

query state $s_q$ and the key state $s_k$. These are passed through a primary encoder $f_{\phi_q}$ and a momentum encoder $f_{\phi_k}$ separately to obtain the corresponding representations $z_q$ and $z_k$. For those pairs, $((z_{q_i}, z_{k_i})$ with the same input and different augmentations is the positive pair, and all other $((z_{q_i}, z_{k_{j \neq i}})$ pairs with different inputs are negative pairs. In the pair $((z_{q_i}, z_{k_i})$, $z_{k_i}$ is considered as a corresponding positive key representation $z_{k_+}$ because it has the same input as the query state $s_q$. Accordingly, its state $s_{k_i}$ is considered as $s_{k_+}$. Simultaneously, a reward model $R_\theta$ evaluates both augmented states to generate reward values $R_\theta(s_q)$ and $R_\theta(s_k)$. These rewards will be used for the KL divergence regularization in the contrastive learning loss equation 4.

## A.9 REWARD SMOOTHNESS ANALYSIS UNDER $\ell_2$ CONSISTENCY REGULARIZER

To investigate the effect of the $\ell_2$ reward consistency regularizer proposed in CIRL, we construct a controlled experiment to measure whether enforcing $R(s) \approx R(s')$ for augmented state pairs $(s, s')$ constrains the reward too strongly. Specifically, we quantify the smoothness of the reward by tracking the **standard deviation of reward differences**:

$$\Delta R = R(s) - R(s'), \qquad \text{Smoothness} = \text{Std}(|\Delta R|).$$

A lower $\text{Std}(|\Delta R|)$ indicates a smoother reward function, meaning that augmented pairs exhibit more consistent reward values. We evaluate four regularization strengths:

$$\beta \in \{0, 10^{-4}, 10^{-3}, 10^{-2}\},$$

while keeping all other CIRL parameters fixed (Adam optimizer, 8-D embedding, and the encoder pretrained using contrastive learning on NGSIM).

### A.9.1 EXPERIMENTAL SETUP

We randomly sample 512 original driving states from NGSIM and apply small semantic augmentations (perturbations in relative position, velocity, TTC, etc.). A small two-layer MLP is trained using InfoNCE loss combined with the $\ell_2$-consistency term:

$$\mathcal{L}_{cont} = \mathcal{L}_{\text{InfoNCE}} + \beta \|R(s) - R(s')\|_2^2.$$

During training, for each $\beta$ we record:

- the InfoNCE contrastive loss over epochs,
- $\text{Std}(|\Delta R|)$, the smoothness of reward differences.

### A.9.2 RESULTS

**Reward Smoothness:** Fig. 10 shows that stronger $\beta$ monotonically reduces the variability of $\Delta R$ across epochs. This confirms that the regularizer successfully makes the reward function smoother.

**Contrastive Representation Quality:** Fig. 11 shows that moderate values $(10^{-3}$–$10^{-2})$ slightly improve $\mathcal{L}_{cont}$ loss training stability, whereas extremely small $\beta$ behaves similar to the baseline $(\beta = 0)$.

### A.9.3 DISCUSSION

Our findings clearly show:

- $\beta = 0$ leads to a noisier reward landscape.
- Very small values $(10^{-4})$ only marginally help.
- $\beta = 10^{-3}$ and $10^{-2}$ produce the most stable reward while preserving contrastive performance.
- The regularizer does not collapse reward differences; rather, it reduces high-frequency noise while maintaining learned structure.

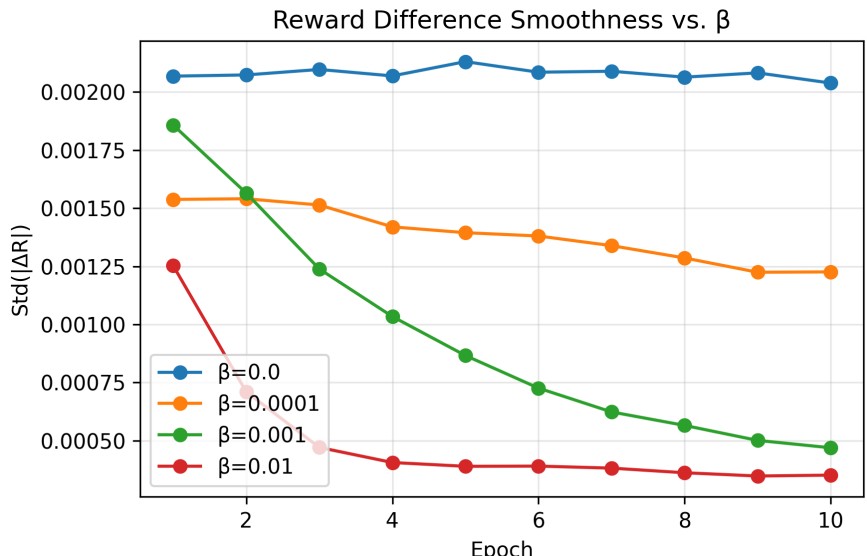

**Figure 10:** Standard deviation of reward difference $|\Delta R|$ across training epochs under different regularizer strengths. Larger $\beta$ enforces smoother reward values.

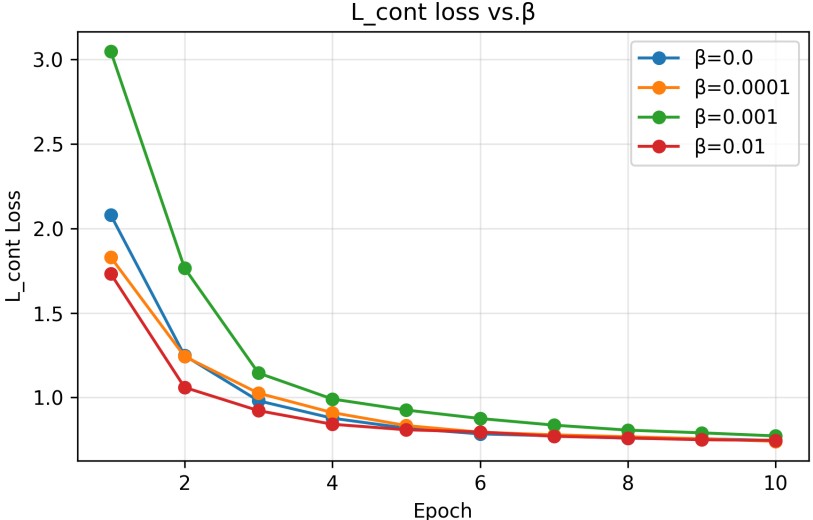

**Figure 11:** $\mathcal{L}_{cont}$ loss across epochs for different $\beta$. Moderate regularization improves training stability without hurting performance.

Overall, the $\ell_2$ consistency regularizer does not interfere with MaxEnt-IRL optimization. Instead, it improves smoothness in a way that better aligns with the geometric structure of the driving state space.

### A.10 T-SNE VISUALIZATION OF CONTRASTIVE FEATURES

To diagnose whether the contrastive encoder learns meaningful behavioral representations, we perform an early-stage visualization using t-SNE on two datasets:(i) expert human trajectories, and (ii) randomly sampled non-human trajectories used during scene exploration.

This analysis is conducted before reward learning and policy optimization, and therefore does not involve any CIRL-generated trajectories.

Fig. 3 illustrates the embedding distribution of 980 expert scenes versus 6000 randomly generated scenes. The expert samples (yellow) form a compact and isolated cluster, whereas random scenes (purple) disperse broadly across the feature space. This clear separation demonstrates that the contrastive encoder successfully captures behavior-discriminative structure and can distinguish human-like trajectories from non-human ones.

For comparison, Figure 12 shows a t-SNE projection of expert scenes and scenes sampled from the CIRL planner after reward learning. Both sets of features occupy similar regions in the embedding space, indicating that the final CIRL reward model encourages the planner to generate behaviors aligned with the expert's feature manifold.

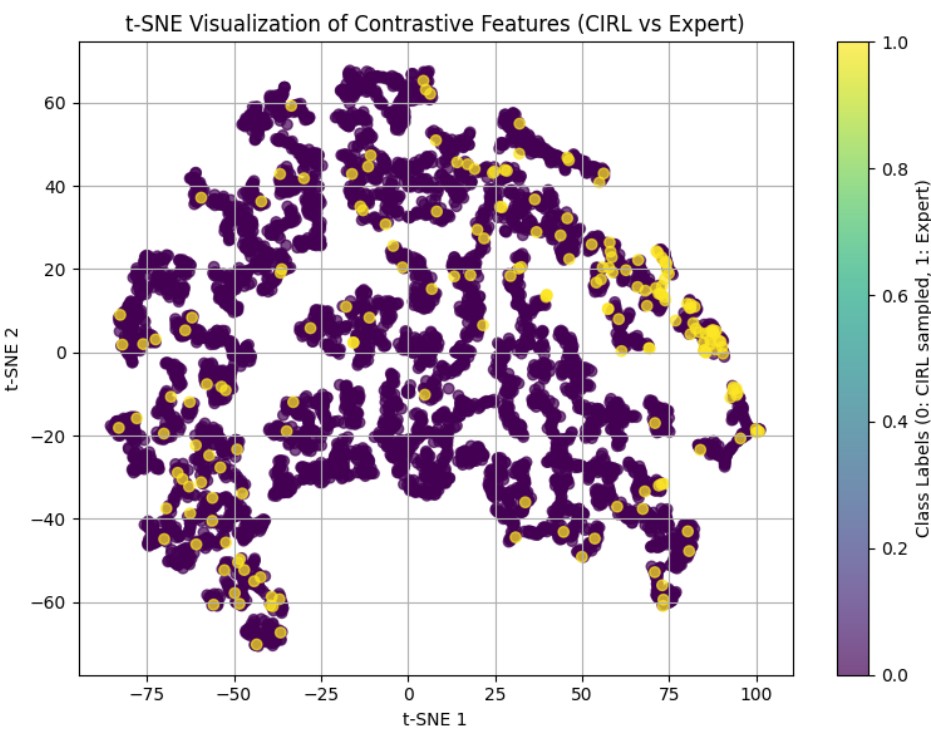

**Figure 12: t-SNE visualization: CIRL vs. expert scenes.** Both CIRL and expert trajectories lie in similar regions of the embedding space, suggesting that the learned reward function successfully drives the policy toward expert-like behavioral representations.

