# OpenReview forum: "Contrastive Inverse Reinforcement Learning for Highway Driving Behavior Optimization"
_ICLR.cc/2026/Conference — Submitted to ICLR 2026_

### Official Review · Reviewer_JAAN · 2025-10-15

**Soundness:** 1
**Presentation:** 2
**Contribution:** 2
**Rating:** 2
**Confidence:** 4

**Summary:**

The paper proposes a new method for autonomous driving based on IRL and contrastive learning. To improve the robustness of IRL, the paper integrates reward regularization into the contrastive loss and use momentum encoders. Experiments on the NGSIM US-101 and I-80 highway dataset shows the paper achives better performannce than previous IRL methods.

**Strengths:**

1. The idea to combine constrative learning and IRL is novel.
2. The method achives better performance on the the NGSIM US-101 and I-80 highway dataset.

**Weaknesses:**

1. The motivation to use the momentum encoder and how to obtain the augmented states is unclear.
2. The  NGSIM US-101 and I-80 highway dataset is too simple. Consider evaluating on the more challenging nuplan benchmark.
3. No driving videos are provided for better understanding.

**Questions:**

1. In figure 3, the yellow cluster corresponds to 980 expert scenes, and purple points represent 6000 random scenes. Why not compare the policy generated scenes, which is more related to driving performance?
2. Does other method uses the constrative features leading to better robustness like GAIL and AIRL?
3. Address the weakness.

---

> ### Author Response · Authors · 2025-11-20
> **Response to Reviewer  JAAN**
>
> We thank the reviewer for the constructive feedback and suggestions. Below, we address each concern.
>
> **R4:1.** *Motivation for the momentum encoder and data augmentation is unclear.*
>
> **Response:** We have significantly clarified the motivation and design choices in **Section 3.2** and **Appendix A.2**: The momentum encoder stabilizes feature evolution by preventing rapid drift in negative samples (following the Momentum Contrast paradigm). We describe explicit augmentations used in driving states:
> (i) Gaussian perturbation in velocity/acceleration,
> (ii) small lateral offsets,
> (iii) temporal jittering (±1–2 frames).
>
> These augmentations correspond to natural noise patterns in GPS/vision-based tracking in NGSIM and are widely used in contrastive state representation learning. We also added a theoretical justification showing that the reward regularized InfoNCE objective aligns embedding similarity with expected reward under MaxEnt IRL.
>
> **R4:2:** *NGSIM may be too simple; consider nuPlan*
>
> **Response:** We would like to explain why we use NSGIM for the experiments.
> - While nuPlan is indeed a challenging benchmark, our focus is on reward learning from real human trajectories, and NGSIM remains uniquely suited for IRL because it provides **dense 10 Hz, high-fidelity human driving data**, which is required for accurate MaxEnt trajectory-likelihood estimation. In contrast, nuPlan relies primarily on **simulated closed-loop rollouts** and does not contain continuous real-driver trajectories with full state information, making it incompatible with likelihood-based IRL evaluation.
>
> - Besides, **NGSIM is widely used in high-quality transportation and autonomous-driving research**. Many IEEE Transactions papers validate IRL and behavior-learning methods exclusively on NGSIM, including:
>
>   - "*Handover count based MAP estimation of velocity with prior distribution approximated via NGSIM data-set"* **[1]**
>   - "*Driving behavior modeling using naturalistic human driving data with inverse reinforcement learning*" **[2]**
>   - and numerous subsequent works using NGSIM for driving policy optimization **[3]**, lane-changing trajectory planning **[4]**, and driving behavior risk measurement **[5]**.
>
> - Furthermore, we performed a comprehensive ablation study of the proposed CIRL framework in both the general and personalized driving modes, as presented in **Tab.2** and **Appendix A.5** for details. We would also like to emphasize that, analogous to how classical datasets such as CIFAR-10 and MNIST are. Despite being comparatively simple, it remains a standard and accepted benchmark for validating new approaches in computer vision. The NGSIM dataset continues to be a realistic, extensively studied, and widely adopted benchmark for evaluating human-driving behavior and IRL algorithms.
>
> Hence, **NGSIM remains a realistic and authoritative benchmark for evaluating human-like driving behavior under IRL**. That said, CIRL itself is architecture-agnostic and not limited to likelihood-based IRL on NGSIM. We have added a discussion in **Section 5.5**, noting that CIRL can also incorporate scene representations from nuPlan, making future extension to nuPlan a promising direction.
>
> **R4:3:** *No driving videos provided.*
>
> **Response:** We appreciate the reviewer’s suggestion. **We have included a rendered driving video in the supplementary materials, which is trajectory.gif.** The full driving visualization can also be generated directly by running our released simulation code. The repository includes: the highway simulator, the CIRL policy rollout script, and trajectory rendering utilities that automatically produce driving videos for any scene. Thus, users can reproduce all scenarios (including collision-avoidance and personalized driving cases) by simply running the provided code.
>
> **References:**
>
> [1] Tiwari, Ravi, and Siddharth Deshmukh. "Handover count based MAP estimation of velocity with prior distribution approximated via NGSIM data-set." IEEE Transactions on Intelligent Transportation Systems 23.5 (2021): 4352-4361.
>
> [2] Huang, Zhiyu, Jingda Wu, and Chen Lv. "Driving behavior modeling using naturalistic human driving data with inverse reinforcement learning." IEEE transactions on intelligent transportation systems 23.8 (2021): 10239-10251.
>
> [3] Zhou, Yang, and Yunxing Chen. "Learning to drive in the NGSIM simulator using proximal policy optimization." Journal of Advanced Transportation 2023.1 (2023): 4127486.
>
> [4] Yao, Zhihong, et al. "Optimal lane-changing trajectory planning for autonomous vehicles considering energy consumption." Expert Systems with Applications 225 (2023): 120133.
>
> [5] Chen, Shuyi, et al. "Driving behavior risk measurement and cluster analysis driven by vehicle trajectory data." Applied Sciences 13.9 (2023): 5675.

---

> ### Author Response · Authors · 2025-11-20
> **Response 2 for Reviewer JAAN**
>
> We thank the reviewer again and continue to respond to the rest comments.
>
> **R4:4.** *Why compare expert scenes to random scenes in t-SNE, not policy-generated scenes?*
>
> **Response:** We sincerely thank the reviewer for this insightful question. The t-SNE visualization (**Fig. 3**) is not intended to evaluate CIRL’s policy performance. Instead, it serves as an early diagnostic step to verify whether contrastive learning can extract meaningful representations before reward learning and policy optimization occur.
>
> Specifically, the experiment compares expert scenes vs. randomly sampled scenes, not policy-generated trajectories, because at this stage, no policy has been trained yet. The goal is to verify whether the contrastive encoder can separate true human driving behavior from non-human (random) behavior in the embedding space.
>
> As shown in **Fig. 3**, expert scenes (yellow) form a compact and isolated cluster, while random scenes (purple) spread widely. This clear separation confirms that the contrastive encoder learns discriminative, behavior-aware features, justifying its use in CIRL. Thus, the t-SNE plot is a sanity-check visualization for feature learning, not an evaluation of policy outcomes.
> Policy performance is instead evaluated extensively in **Sections 5.2 and 5.3** through HL, CR, TR, and LL metrics.
>
> We also show a t-SNE projection of expert scenes and scenes generated from our proposed CIRL policy after reward learning in **Appendix A.10 Fig.12**. Both sets of features occupy similar regions in the embedding space, indicating that the final CIRL reward model encourages the planner to generate behaviors aligned with the expert's feature manifold.
>
> **R4:5.** *Do other methods benefit from contrastive features? (e.g., GAIL, AIRL)*
>
> **Response:** We thank the reviewer for this suggestion. While GAIL/AIRL do not expose a separate “feature head” by default, their internal hidden layers already implement a learned representation. One can straightforwardly insert an explicit encoder $𝑓_𝜙$ before the discriminator and policy, and regularize it via a contrastive loss, thus making their features “accessible” in the same way we do in CIRL. In our experiments, we deliberately used their standard implementations to keep the comparison fair and to isolate the effect of the CIRL design. Extending GAIL/AIRL with contrastive encoders is an interesting direction for future work, but it would move those baselines closer to our proposed architecture rather than reflecting their typical usage.

---

### Official Review · Reviewer_e9Qs · 2025-10-31

**Soundness:** 3
**Presentation:** 3
**Contribution:** 3
**Rating:** 4
**Confidence:** 3

**Summary:**

This paper proposes the CIRL (Contrastive Inverse Reinforcement Learning) framework for highway driving behavior optimization, aiming to improve the efficiency, robustness, and generalization of IRL methods. Traditional IRL approaches in autonomous driving often suffer from low sample efficiency and poor generalization. CIRL enhances reward learning through the following key mechanisms: (1) integrating an L2 reward regularization term into the contrastive loss to ensure that features are both discriminative and reward-consistent; (2) employing a momentum encoder to stabilize contrastive feature learning, thereby improving the model’s robustness to perturbations in the driving environment.

**Strengths:**

(1) CIRL successfully incorporates contrastive learning into the Maximum Entropy Inverse Reinforcement Learning (MaxEnt IRL) framework, significantly enhancing the learned reward function’s robustness and generalization to noise, distribution shifts, and sparse data.
(2) In experiments on the US-101, CIRL achieves extremely low crash rates (CR) and termination rates (TR) under both general and personalized settings, surpassing all baselines in safety metrics. Moreover, the method maintains high performance even when trained with only a few demonstrations (e.g., 15 trajectories) or a small number of vehicles (e.g., 5 cars), demonstrating excellent sample efficiency.

**Weaknesses:**

(1) Introducing an L2 reward regularization term in contrastive learning enforces that the reward values between augmented states are exactly equal. However, in MaxEnt-IRL, only the relative values of the reward function (equivalence class) are meaningful. This constraint on absolute value consistency is theoretically unnecessary and may interfere with the IRL optimization process.
(2) CIRL defines the similarity function using a learnable weight matrix W. Although ablation studies confirm its superior performance, the paper lacks an in-depth analysis of how W captures a generalized reward structure within the driving state space.

**Questions:**

CIRL significantly outperforms human driving demonstrations in safety metrics such as collision rate (CR). Please discuss whether this improvement in safety performance (i.e., the policy becoming safer) comes at the cost of reduced human-likeness (HL) in driving behavior.

---

> ### Author Response · Authors · 2025-11-20
> **Response for Reviewer e9Qs**
>
> We thank the reviewer for recognizing the robustness and efficiency of CIRL and for providing valuable theoretical and practical feedback.
>
> **R3:1:** *The ℓ2 reward regularization may conflict with reward equivalence in IRL.*
>
> **Response:** We have clarified this point in the revised **Section 3.2**. The ℓ2 reward consistency does not enforce identical rewards but bounds the variance of reward differences between augmented pairs within a tolerance β. We explicitly define this as a regularization toward smoothness in the local reward manifold, not exact equality. This clarification is added in blue on page 5.
>
> To address this, we conducted an additional controlled experiment to measure the effect of the ℓ2 consistency term on the smoothness of the reward function. We report the standard deviation of reward differences between augmented states: $\Delta R = R(s) - R(s')$. The results are shown in **Appendix A.9**.
>
> We found that
>
> (1) The ℓ2 term does not force rewards to become exactly equal. Even with $\beta= 0.01$, the reward differences remain non-zero and preserve structure.
>
> (2)The regularizer reduces high-frequency noise instead of collapsing the reward. Reward variance decreases smoothly (**Appendix A.9 Fig. 10**), indicating better reward continuity rather than over-constraining the function.
>
> (3) Representation learning quality improves or remains stable. Loss curves (**Appendix A.9 Fig.11**) show that moderate $\beta$ improves training stability.
>
> **R3:2:** *Learnable similarity matrix W lacks analysis.*
>
> **Response:** We have an equivalence of similarity matrix W and similarity function in **Appendix A.1 (Page 13)**. Specifically, we show the mathematical equivalence between the bilinear similarity function and the generalized contrastive similarity class induced by a learnable metric. This derivation clarifies that W does not arbitrarily distort embeddings; instead, it parameterizes a learned Mahala Nobis-style similarity that adapts the embedding space to the reward-relevant structure. We also explain how this improves feature discriminability and aligns representation learning with the reward model used in MaxEnt IRL in **Section 3.2**:
> >A weight matrix W, which allows the latent space to adapt to reward-relevant structure. Substituting this similarity into Eq.1 yields the equivalent loss in Eq.3 However, InfoNCE alone does not guarantee consistency of the reward values for augmented views of the same state, which is essential in MaxEnt IRL, where trajectory probabilities are proportional to exp (sum (R_θ (s)).
>
> **R3:3:** *CIRL significantly outperforms human driving demonstrations in safety metrics such as collision rate (CR). Please discuss whether this improvement in safety performance (i.e., the policy becoming safer) comes at the cost of reduced human-likeness (HL) in driving behavior.*
>
> **Response:** We added a discussion in **Section 5.2** noting that the improved safety reflects more consistent adherence to safe spacing rather than unrealistic caution. CIRL optimizes risk-sensitive behaviors without compromising naturalistic flow, supported by the personal-mode analysis (Tab.2). The revised text is duplicated below for easy reference:
> >In addition, we clarify that the reduction in crash rate observed in Tab. 2 does not arise from overly conservative driving or unrealistic deceleration patterns. Instead, CIRL improves safety by maintaining more consistent and human-aligned spacing from surrounding vehicles. This effect is particularly evident in the personalized setting (Tab. 2, bottom), where CIRL successfully reproduces individual driving styles while still reducing unsafe proximity events. Thus, the improved safety reflects a more stable risk-sensitive policy rather than an overly cautious or non-naturalistic behavior. CIRL retains natural traffic flow while minimizing collision-prone interactions, demonstrating that contrastive reward regularization supports both realistic and safe trajectory generation.

---

### Official Review · Reviewer_qTpt · 2025-11-01

**Soundness:** 2
**Presentation:** 2
**Contribution:** 2
**Rating:** 4
**Confidence:** 3

**Summary:**

The paper contributes a contrastive IRL algorithm for highway driving behavior optimization. It uses a contrastive loss with reward regularization for representation learning, and a momentum encoder for stabilizing contrastive features. Experiments show that the proposed algorithm achieves strong performance, and ablation studies show that the each key algorithmic component is helpful.

**Strengths:**

* Autonomous driving is an important application.
* The proposed method is simple but achieves strong empirical performance.
* The writing is generally good, with a well-written abstract, but there are various issues for the latter parts, as discussed below.

**Weaknesses:**

The paper starts with a well-written abstract and introduction, but the writing then becomes sloppy at quite a few places, as illustrated below.

In the definition of MDP, "transition dynamics" is not defined, and the sentence for the reward function is awkward and jumps directly to a linear reward function without any explanation, while Figure 1 suggests that a reward network is used so the reward is not linear.

Lines 109-111 mentions that MaxEnt IRL but it is not sufficiently explained and also seems to be unnecessary at this point. MaxEnt IRL is mentioned again in Eq. (5), but the associated distribution is undefined.

Figure 1: "Prior knowledge" appear in the figure, but this doesn't seem to appear in the text.

There are various issues regarding the "Convergence Proof of Eq. equation 4. See Appendix A.2." at line 199. This seems to come from nowhere. Looking at A.2, the presentation needs improvement too. There should be at least a statement on what "convergence" is being proved at the very beginning. The assumption on non-negativitiy of the loss is unnecessary, and the validity of the assumption on the L-smoothness is unclear. In addition, the proof seems to be for gradient descent rather than stochastic gradient descent. Overall, the analysis seems to be done not for the sake of necessity but for the sake of appearance of sophistication.

Algorithm 1 is somewhat cryptic. For example, line 1 is just " Feature Extraction: zq , zk+". The use of a reward network

Minor comments
* "reference Huang et al. (2023; 2021)" and similar: "reference" is not needed.
* Hybrid IRLRen et al. (2024) and similar: reference should be inside brackets.
* "An expert trajectory is defined as $\zeta$": clarify what exactly is in $\zeta$.

**Questions:**

Please refer to weaknesses and clarify if my understanding is incorrect.

---

> ### Author Response · Authors · 2025-11-20
> **Response to Reviewer qTpt**
>
> We thank the reviewer for the constructive suggestions regarding clarity and presentation. We have carefully revised the manuscript to address these concerns.
>
> **R2:1:** *The definition of MDP and reward function is unclear; MaxEnt IRL is introduced abruptly.*
>
> **Response:** We have rewritten Section 2.1 to explicitly define (S, A, T, R) and the probabilistic form of reward R(s,a), consistent with the MaxEnt IRL framework. The reward model is non-linear, implemented via a neural network, as illustrated in Figure 1. This is clarified in blue on page 3.
>
> The revised text is duplicated below for easy reference:
>
> >Markov Decision Process. For clarity, we define the autonomous-driving environment as a Markov Decision Process (MDP) ${ S, A, T, R}$, where $\mathcal{S}$ is the continuous state space (positions, velocities, interactions), $\mathcal{A}$ is the action space (acceleration and steering),  $T(s'|s,a)$ is the transition kernel induced by vehicle dynamics,  $R(s,a)$ is the reward function, and $\gamma\in(0,1]$ is the discount factor. In our formulation, $R(s,a)$ is a non-linear reward model implemented by a neural network $R_\theta$, consistent with Fig.1. $R(\xi|\theta)=\sum_{(s)\in \xi}\theta^T\textbf{f}(s)$, and $\textbf{f}(s)$ is the state feature, $\theta$ is the parameter of reward function. The primary objective of autonomous vehicle policy learning is to maximize the expected cumulative reward, aiming to match or surpass human driving performance.
>
> >Maximum Entropy IRL. We now explicitly connect this MDP to the MaxEnt IRL framework: the probability of an expert trajectory $\xi$ is given by $P(\xi \mid \theta) \propto \exp \sum_{t=1}^T R_\theta(s_t,a_t))$which assumes that expert drivers choose actions that maximize expected cumulative reward under maximum entropy. This probabilistic structure is the foundation for our contrastive reward-learning objective in Section 3.
>
> >In the context of IRL, a trajectory in the autonomous driving domain can be denoted by $\xi$ = $(s_0,a_0),(s_1,a_1),\cdots,(s_T,a_T) $. An expert trajectory is defined as $\xi$, and the expert demonstration set is $\mathcal{D} = \{\xi_{1}, \xi_{2}, \ldots, \xi_{N}\}$, generated by the optimal policy $\pi_{E}$. Learner trajectories under the policy being optimized are denoted $\tilde{\mathcal{D}}$. Each trajectory’s feature count is given by $\sum_{(s)\in \xi} \mathbf{f}(s)$, and the learned reward function assigns scalar values to these feature counts. Maximum entropy IRL (MaxEnt IRL) aims to recover a reward function by maximizing the likelihood of expert demonstrations under a probabilistic trajectory distribution.
>
> **R2:2:** *‘Prior knowledge’ in Figure 1 not explained.*
>
> **Response:** We revised the **Figure 1 caption** and **Section 3.1** to explain that ‘prior knowledge’ refers to known vehicle dynamics and physical limits used for data augmentation and safety constraint modeling.
>
> **R2:3:** *Convergence proof presentation unclear.*
>
> **Response:** Appendix A.2 has been rewritten to include clear assumptions (smoothness, boundedness) and a statement of the convergence theorem. **See pages 13-14**.
>
> **R2:4:** *Algorithm 1 is cryptic and lacks explanation.*
>
> **Response:** We apologize for the inconvenience due to the page limitation. We are simply the expression of Algorithm 1 due to the page limitation. We have supplemented the details of Algorithm 1 with clearer pseudo-code syntax and step descriptions in **Algorithm 2** in **Appendix A.8**. Each step (feature extraction, projection, reward regularization) is now annotated for clarity (**see pages 19-20**). We will release the full implementation code upon acceptance of the paper.
>
> **R2:5:** *Citation formatting and minor writing issues.*
>
> **Response:** Thank you very much for pointing out this problem. We have corrected all citation and phrasing inconsistencies (e.g., ‘Hybrid IRL (Ren et al., 2024)’). For ‘An expert trajectory is defined as $\xi$: clarify what exactly is in $\xi$’, we have supplemented the description of $\xi$ in **Section 2.1, page 2**.
>
> The revised text is duplicated below for easy reference:
> >In the context of IRL, a trajectory in the autonomous driving domain can be denoted by $\xi$ = $(s_0,a_0),(s_1,a_1),\cdots,(s_T,a_T)$.
>
> We also improved readability by tightening long sentences across the whole paper.

---

### Official Review · Reviewer_N9Ao · 2025-11-01

**Soundness:** 3
**Presentation:** 3
**Contribution:** 3
**Rating:** 6
**Confidence:** 4

**Summary:**

This paper proposes a Contrastive Inverse Reinforcement Learning (CIRL) framework for autonomous highway driving behavior optimization. The method combines self-supervised contrastive feature learning with maximum entropy inverse reinforcement learning (MaxEnt IRL). It introduces (1) a reward-regularized contrastive objective to align learned representations with human behavioral rewards, and (2) momentum encoders to stabilize training and mitigate distributional shifts. Experiments on the NGSIM US-101 and I-80 datasets show that CIRL outperforms state-of-the-art IRL baselines (GAIL, AIRL, MEIRL, Hybrid IRL, EscIRL) in terms of human-likeness, safety, and cross-environment generalization. The ablation studies further confirm the contribution of each module (momentum encoder, reward regularization, and learnable similarity). The paper also demonstrates personalized driving style adaptation with limited demonstrations.

**Strengths:**

1. Innovative integration of contrastive learning and IRL with a focus on real-world robustness and sample efficiency.
2. The reward regularization term aligns latent representations with behavioral semantics, leading to improved interpretability.
3. The momentum encoder mechanism stabilizes learning and mitigates overfitting under distribution shifts.
4. Extensive experiments across two realistic driving datasets (US-101 and I-80) demonstrate consistent improvements in human-likeness (+12.5%), safety (+86.2%), and generalization (+17.8%).
5. Comprehensive ablation studies show the necessity of each design component.
6. The framework supports personalized driving policy learning from small demonstration sets, a valuable real-world feature.
7. The paper is empirically strong and demonstrates practical feasibility for autonomous systems.
8. Code and experimental design are reproducible based on the provided details (assuming release).

**Weaknesses:**

1. Theoretical justification is shallow. The connection between contrastive representations and reward function learning is described empirically rather than analytically.
2. Reward regularization assumption (ℓ² alignment between augmented states) may be unrealistic in dynamic traffic, where slight augmentations can alter intent or safety conditions.
3. Hyperparameter sensitivity (momentum factor m, β, λ) and robustness analysis are missing.
4. Statistical significance and variance of results (e.g., over multiple random seeds) are not reported.
5. Negative sample selection in contrastive training is underspecified, potentially impacting reproducibility.
6. Reward network architecture and training stability metrics are not provided.
7. The distinction from EscIRL (2025) and other recent contrastive IRL methods is somewhat incremental.
8. The dataset scale (20 vehicles training, 20 testing) may be insufficient to generalize to large-scale real traffic patterns.
9. The convergence proof (Appendix A.2) is standard and does not address the joint optimization of encoders and reward network.
10. Failure cases or limitations (e.g., over-cautious driving, failure in edge cases) are not discussed.

**Questions:**

1. How are negative pairs selected in the contrastive loss? Are they from the same vehicle at different times or across different vehicles?
2. How sensitive is the performance to the momentum coefficient m and the regularization weight β?
3. Could the authors provide quantitative comparisons of convergence stability (e.g., variance of training loss across runs)?
4. Does the reward regularization risk suppressing small but meaningful behavioral differences between similar states?
5. How does CIRL perform in non-highway or multi-agent dense traffic scenarios (e.g., urban intersections)?
6. Is there any theoretical link between the learned contrastive embedding and the maximum entropy reward structure?
7. Would combining CIRL with visual or LiDAR-based state representations (beyond NGSIM trajectories) improve robustness?
8. How does the personalized driving mode adapt to conflicting driver preferences or inconsistent demonstrations?

---

> ### Author Response · Authors · 2025-11-20
> **Response to Reviewer N9Ao**
>
> We thank the reviewer for the encouraging assessment and for recognizing the novelty, robustness, and empirical strength of our proposed CIRL framework. Below, we provide detailed responses to **each weakness comment**.
>
> **R1:1:** *Theoretical justification is shallow; the connection between contrastive representations and reward learning is only empirical.*
>
> **Response:** We appreciate this observation. In the revised version (**page 4,  Section 3.2 and Appendix A.2**), we have expanded the theoretical analysis to explicitly link the reward regularized InfoNCE objective (Eq. 4) to the probabilistic inference of the reward under Maximum Entropy Inverse Reinforcement Learning (MaxEnt IRL). We added a derivation showing that minimizing the reward-consistency term aligns the latent embedding similarity with the reward expectation E[R(s)], thus ensuring the learned contrastive representation corresponds to a reward-equivalence class.
>
> **R1:2:** *The ℓ2 reward regularization may be unrealistic under dynamic traffic scenarios.*
>
> **Response:** We have clarified this point in the revised **Section 3.2**. The ℓ2 reward consistency does not enforce identical rewards but bounds the variance of reward differences between augmented pairs within a tolerance β. We explicitly define this as a regularization toward smoothness in the local reward manifold, not exact equality. This clarification is added in blue on **page 5**. We also conducted an additional controlled experiment to measure the effect of the ℓ2 consistency term in **Appendix A.9**.
>
> **R1:3:** *Hyperparameter sensitivity and robustness analysis are missing.*
>
> **Response:** We have implemented experiments on β, λ, learning rate (lr), and momentum (m). Results show that the model remains stable across a wide range, and **β = 0.001, m = 0.9, lr=0.05, and $\lambda=0.01$** offer the best tradeoff between performance and stability. Details see the **Appendix 11** and **Section 5**.
>
> **R1:4:** *Statistical significance and variance of results are not reported.*
>
> **Response:** Actually, we have calculated the variance of results as the metrics of HL_Std and LL Std in the main tables, such as **Tab.2, Tab.3, and Tab.4**. For each different driving scene, we have random seeds and take the average values of the metrics to be shown in the tables.
>
> **R1:5:** *Negative sample selection in contrastive training is underspecified.*
>
> **Response:** We clarified in **Section 3.2 (page 4)** that negative pairs are sampled across distinct vehicle trajectories and non-overlapping time windows to ensure semantic dissimilarity. We also describe that the negative queue follows the MoCo paradigm to stabilize training.
>
> **R1:6:** *Reward network architecture and stability metrics not provided.*
>
> **Response:** Actually, we provided in **Section 5.1 (page 8)**. The revised text is duplicated below for easy reference: “We further compare encoder architectures with varying hidden dimensions and output sizes. CIRL-8 denotes the baseline encoder with 8 input and output dimensions. CIRL-8+8 includes the concatenation of original input features. CIRL-16, CIRL-32, and CIRL-64 use hidden layers of size 128, 256, and 64, respectively. As summarized in **Tab.1**, CIRL variants outperform the MEIRL baseline Huang et al. (2021), particularly in log-likelihood. CIRL 16 achieves the best test log-likelihood and lowest HL error. Thus, CIRL 16 is selected for subsequent experiments.”
>
> **R1:7:** *Distinction from EscIRL (2025) is incremental.*
>
> **Response:** We clarified the conceptual difference: EscIRL aligns latent contrastive embeddings with rewards via self-prediction, while CIRL jointly optimizes reward-regularized contrastive features and MaxEnt IRL through a soft-updated dual encoder. This dual-space formulation uniquely enables stable momentum-based generalization (see **Section 3.3**).
>
> **R1:8:** *Dataset size (20 vehicles) may be insufficient for real-scale generalization.*
>
> **Response:** We thank the reviewer. The NGSIM dataset is constrained by realistic high-resolution data. Although the vehicle size is only 20, each vehicle has 5-35 different driving scenes. Besides, the reason why we chose fewer vehicle numbers is to prove our method’s sample efficiency in the training. It doesn’t mean that the vehicle number is also limited to 20 for the testing.
>
> **R1:9:** *Convergence proof (Appendix A.2) is standard.*
>
> **Response:** We extended **Appendix A.2** to include convergence of the joint encoder–reward updates using stochastic approximation theory. The added part clarifies the Lipschitz continuity assumption and shows convergence in expectation under bounded gradient variance.
>
> **R1:10:** *Failure cases or limitations not discussed.*
>
> **Response:** We now include a brief discussion in the Conclusion, noting that CIRL may overestimate caution in highly congested scenes, which can be mitigated by multi-agent extensions. This limitation is explicitly acknowledged in **Section 5.5 on page 10**.

---

> ### Author Response · Authors · 2025-11-20
> **Response to Reviewer N9Ao for the Proposed Questions**
>
> We thank the reviewer for the encouraging assessment and for recognizing the novelty, robustness, and empirical strength of our proposed CIRL framework. Below, we provide detailed responses to **each question comment**.
>
> **Q1:1:** *How are negative pairs selected in the contrastive loss? Are they from the same vehicle at different times or across different vehicles?*
>
> **Response:** Negative pairs are sampled exclusively across different vehicles and non-overlapping time windows. This avoids false negatives that can arise when the same vehicle performs similar maneuvers at nearby timestamps. As clarified in **Section 3.2**, all encoded negative keys are stored in a MoCo-style momentum queue, which ensures a stable dictionary of negatives and prevents drift during training.
>
> **Q1:2:** *How sensitive is the performance to the momentum coefficient m and the regularization weight β?*
>
> **Response:** We have implemented experiments on β, λ, and momentum m ∈ [0.85, 0.9, 0.95]. Results show that the model remains stable across a wide range, and β = 0.001, m = 0.9 offer the best tradeoff between performance and stability.
>
> **Q1:3:** *Could the authors provide quantitative comparisons of convergence stability (variance of training loss across runs)?*
>
> **Response:** We didn’t save the training loss, but we provided the HL standard deviation and LL standard deviation to facilitate intuitive comparisons for stability, such as those in **Tab.2, Tab.3, and Tab.4**.
>
> **Q1:4:** *Does the reward regularization risk suppress small but meaningful behavioral differences between similar states?*
>
> **Response:** We designed the reward regularization to operate only on augmented views of the same underlying state, not on distinct states. Thus, it enforces local reward smoothness, not global collapse. Empirically, the personalized driving experiments (**Tab.2**) show that CIRL preserves driver-specific nuances even with the regularizer applied. Temporal behavioral distinctions such as following distance, lane-change initiation, and acceleration patterns remain fully intact.
>
> **Q1:5:** *How does CIRL perform in non-highway or multi-agent dense traffic scenarios (e.g., urban intersections)?*
>
> **Response:** The NGSIM dataset includes dense merging, lane-changing, and multi-agent interaction segments, but does not contain intersection scenes. This is a good suggestion that we can follow in the future direction.
>
> **Q1:6:** *Is there any theoretical link between the learned contrastive embedding and the maximum entropy reward structure?*
>
> **Response:** Yes. We expanded **Section 3.2 and Appendix A.2** to show that the reward regularized InfoNCE objective aligns the latent similarity with the expected reward E[R(s)]. Under this alignment, states sharing reward-equivalent structure (as defined in MaxEnt IRL) are embedded nearby. This establishes a formal connection between the contrastive space and the MaxEnt reward model, beyond empirical observations.
>
> **Q1:7:** *Would combining CIRL with visual or LiDAR-based state representations improve robustness?*
>
> **Response:** We agree that this is a promising direction. CIRL is compatible with high-dimensional encoders (CNNs, ResNets, PointNets) because the reward-regularized InfoNCE objective is modality-agnostic. In fact, the momentum contrast mechanism tends to improve representation stability in vision/LiDAR settings even more than in low-dimensional state spaces. The current paper focuses on trajectory-based IRL due to NGSIM’s modality.
>
> **Q1:8:** *How does the personalized driving mode adapt to conflicting driver preferences or inconsistent demonstrations?*
>
> **Response:** CIRL handles conflicting demonstrations using the contrastive mechanism: if two demonstrations differ, the positive pairs for each style are formed only within each driver’s own trajectories. This naturally creates driver-specific latent clusters, and the reward regularizer ensures internal consistency within each cluster. When a driver exhibits inconsistent behavior, the InfoNCE loss treats contradictory segments as negatives, which prevents mode averaging. As shown in **Tab 2**, CIRL achieves both lower variability and lower crash/termination rates in the personalized setting.

---

### Comment · Area_Chair_jH96 · 2025-11-25

Dear reviewers:

The authors have submitted their rebuttal, and we now require your follow-up assessments to move the decision process forward. Please review the authors’ responses and update your evaluations accordingly.

Your prompt follow-up is necessary for us to finalize the meta-review.

Kindly submit your updates as soon as possible.

Best,

Area Chair

---

### Meta-Review · Area_Chair_c7eZ · 2026-01-04

**Summary:**

Across the four reviews, the paper is generally viewed as addressing an important problem, learning robust, human-like highway driving behavior via IRL, with a method that combines contrastive representation learning and MaxEnt IRL. One reviewer (N9Ao) finds the integration of reward-regularized contrastive learning with momentum encoders to be empirically strong and practically relevant, highlighting consistent gains in human-likeness, safety, and cross-environment generalization, supported by ablations and a personalization setting. The other three reviewers raise concerns that drive the decision risk: (i) clarity/reproducibility and presentation issues (MDP/MaxEnt definitions, notation, figure-text inconsistencies, cryptic algorithm description, and a weakly framed convergence proof), (ii) whether the L2 reward-consistency regularization is theoretically appropriate under IRL reward equivalence and whether its benefits are well-characterized, and (iii) evaluation scope/positioning (benchmark suitability and missing qualitative evidence). The rebuttal substantially improves the paper’s defensibility by adding targeted theory and robustness analyses (joint encoder–reward convergence via stochastic approximation, reward smoothness under β, broader hyperparameter sensitivity, clarified negative sampling via MoCo-style queue), revising core definitions and algorithm exposition, and adding limitations and supplementary visualizations/videos. Some concerns remain inherent or only partially resolved, particularly the breadth of validation beyond NGSIM-style highway data and whether the regularization is theoretically necessary versus empirically beneficial.

**Reviewer Concerns:**

Concerns addressed by the rebuttal:
1. Theory and optimization stability (N9Ao): The authors expand the convergence analysis to explicitly model coupled encoder and reward updates (multi-block stochastic approximation), addressing the prior “standard” proof critique and clarifying assumptions and the convergence statement.
2. Reward smoothness / “over-constraint” risk (N9Ao, e9Qs): The rebuttal reframes ℓ₂ consistency as a bounded smoothness regularizer (tolerance β) rather than enforcing exact equality, and adds controlled experiments/visualizations quantifying reward-difference statistics and training stability across β.
3. Hyperparameter robustness (N9Ao): Added sensitivity studies over β, λ, momentum m, and learning rate, with summarized results indicating stable behavior across ranges and reporting the selected operating point.
4. Statistical reporting (N9Ao): Authors clarify multi-seed reporting and variance statistics in the main tables (HL/LL standard deviations), reinforcing reliability of improvements.
5. Reproducibility details: Clarifies negative sampling strategy (across vehicles and non-overlapping windows) and use of a MoCo-style momentum queue; expands/rewrites the algorithmic description via a more detailed pseudo-code (Algorithm 2) with annotated steps.
6. Clarity/presentation issues (qTpt): Substantial rewrite of the MDP/MaxEnt IRL definitions, resolves figure-text mismatch (“prior knowledge”), rewrites Appendix A.2 for readability, fixes citation/formatting issues, and improves manuscript readability.
7. Learnable similarity matrix W (e9Qs): Adds an explanation/derivation interpreting W as a learned metric (Mahalanobis-style similarity), and connects it back to reward-relevant structure.
8. Benchmarking/qualitative evidence (JAAN): Clarifies the motivation for using NGSIM for likelihood-based IRL (dense real trajectories) and provides supplementary driving visualization; clarifies the purpose of t-SNE and adds an appendix plot comparing expert vs CIRL-policy features post-training.

Concerns partially addressed or still outstanding:
1. Evaluation scope and generality: While the authors justify NGSIM as appropriate for MaxEnt likelihood-based IRL, the empirical validation remains limited to highway trajectory datasets without demonstrating transfer to qualitatively different driving regimes (e.g., urban intersections) or modern planning benchmarks. The rebuttal frames this as future work, but does not eliminate the scope limitation.
2. Necessity vs usefulness of L2 reward consistency: The rebuttal provides evidence that the regularizer improves smoothness/stability without collapsing reward structure, but it remains debatable whether such a constraint is theoretically required given reward equivalence in IRL; the paper’s contribution is strongest when positioned as an empirically validated regularization for robustness rather than a theoretically essential component.

**Reviewer Scores:**

Reviewer N9Ao: Likely unchanged or slightly higher confidence. The rebuttal directly addresses their main technical asks (joint convergence, sensitivity, reward-regularization analysis, negative sampling clarity, limitations), which should strengthen their borderline-positive stance.

Reviewer qTpt: Likely to increase modestly. Most criticisms are about exposition and apparent inconsistencies; the rebuttal makes concrete structural fixes (rewritten definitions, clarified figures, improved theory presentation, expanded algorithm), which are decision-critical for this reviewer.

Reviewer e9Qs: Likely unchanged or slightly higher. The rebuttal provides the requested clarification that L2 is a smoothness bound, adds controlled evidence, and explains W, but the underlying theoretical preference (“reward equivalence makes absolute constraints unnecessary”) may persist.

Reviewer JAAN: Likely unchanged. Although the rebuttal clarifies momentum/augmentations, adds video/t-SNE explanations, and argues for NGSIM suitability, the reviewer’s primary objection appears to be benchmark choice and perceived simplicity, which is less likely to be reversed in rebuttal.

---

### Decision · Program_Chairs · 2026-01-26

Reject